# BitStack: Any-Size Compression of Large Language Models in Variable Memory Environments

**Xinghao Wang**[1], **Pengyu Wang**[1], **Bo Wang**[1], **Dong Zhang**[1], **Yunhua Zhou**[2*], **Xipeng Qiu**[1*]

[1]Fudan University, [2]Shanghai Artificial Intelligence Laboratory

`{xinghaowang22, pywang24, bwang22, dongzhang22}@m.fudan.edu.cn`
`zhouyunhua@pjlab.org.cn, xpqiu@fudan.edu.cn`

## Abstract

Large language models (LLMs) have revolutionized numerous applications, yet their deployment remains challenged by memory constraints on local devices. While scaling laws have enhanced LLM capabilities, the primary bottleneck has shifted from *capability* to *availability*, emphasizing the need for efficient memory management. Traditional compression methods, such as quantization, often require predefined compression ratios and separate compression processes for each setting, complicating deployment in variable memory environments. In this paper, we introduce **BitStack**, a novel, training-free weight compression approach that enables megabyte-level trade-offs between memory usage and model performance. By leveraging weight decomposition, BitStack can dynamically adjust the model size with minimal transmission between running memory and storage devices. Our approach iteratively decomposes weight matrices while considering the significance of each parameter, resulting in an approximately 1-bit per parameter residual block in each decomposition iteration. These blocks are sorted and stacked in storage as basic transmission units, with different quantities loaded based on current memory availability. Extensive experiments across a wide range of tasks demonstrate that, despite offering fine-grained size control, BitStack consistently matches or surpasses strong quantization baselines, particularly at extreme compression ratios. To the best of our knowledge, this is the first decomposition-based method that effectively bridges the gap to practical compression techniques like quantization. Code is available at `https://github.com/xinghaow99/BitStack`.

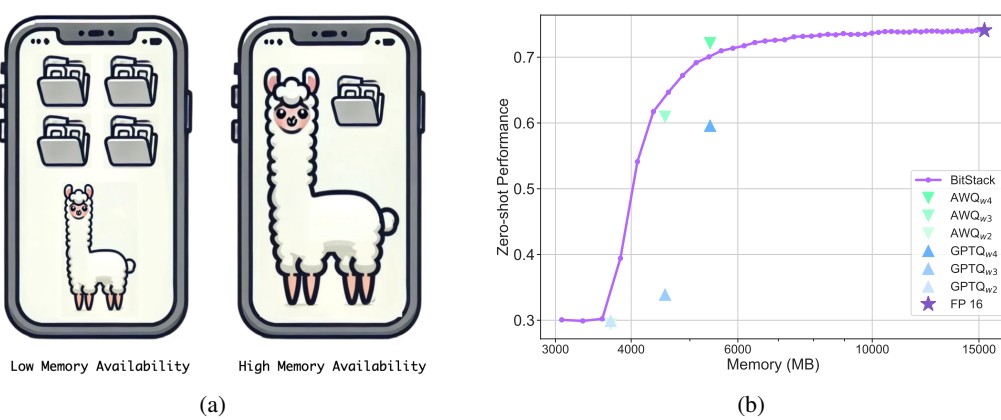

(a)               (b)

Figure 1: **BitStack** enables dynamic compression of LLMs in variable memory environments (a), while still matching or surpassing the performance of practical compression methods such as GPTQ (Frantar et al., 2022) and AWQ (Lin et al., 2024) with the same memory footprint(b).

---

*Corresponding Authors

## 1 INTRODUCTION

Large language models (LLMs) have demonstrated superior performance on various benchmarks (Achiam et al., 2023; Dubey et al., 2024) and are increasingly serving as practical assistants in people's daily lives, such as general language assistants (OpenAI, 2024; Google, 2024; Anthropic, 2024), search engines (Perplexity.AI, 2024), and code assistants (GitHub, 2024).

With the blessing of scaling laws (Kaplan et al., 2020), LLMs are becoming more powerful as their sizes expand, and the main bottleneck for deploying task-capable LLMs has shifted from their *capability* to their *availability*. For example, loading only the weights of the Llama 3.1 8B model (Dubey et al., 2024) requires approximately 14.96 GB of RAM in FP16, not including the activations, which also consume significant memory during inference, especially for long-context tasks.

To adapt to various memory and device constraints, numerous methods have been proposed for LLM compression, such as quantization (Frantar et al., 2022; Lin et al., 2024; Shao et al., 2023; Egiazarian et al., 2024; Tseng et al., 2024), pruning (Ma et al., 2023; Xia et al., 2023; Ashkboos et al., 2024), and distillation (Muralidharan et al., 2024). These methods often compress models to a predefined compression ratio (e.g., specifying numerical precision, defining target structures for pruned models or student models) and require running the compression procedure from scratch for every compression setting. Another line of research for compressing LLMs is weight decomposition (Hsu et al., 2022; Yuan et al., 2023; Wang et al., 2024). These methods compress the model weights via low-rank decomposition but often suffer from severe performance degradation at high compression ratios.

Deploying large language models locally(e.g. on personal computers or smartphones) is a common practice, as it safeguards private data and enables offline functionality. However, the available RAM on these devices is often limited and variable, as the total memory capacity is generally small and memory usage by other applications can fluctuate(Figure 1a). This variability in available memory poses a challenge for deploying LLMs, as they require consistent and substantial RAM resources. For example, when more memory becomes available from other applications, users may want to use a 4-bit quantized model instead of a 3-bit one for better performance. However, this requires reloading the entire model, which may cause significant delays due to limited transmission bandwidth. Additionally, multiple versions of the model at different compression ratios need to be stored on the device, and each version requires running a separate compression process in advance, which increases the storage burden on the device and requires additional computational resources to run separate compression processes. Therefore, a compression strategy that enables dynamic trade-offs between memory usage and performance is highly desirable.

As discussed earlier, achieving these trade-offs requires avoiding compressing towards a fixed ratio. Instead, we aim to compress the model once, allowing it to be dynamically loaded within any arbitrary memory budget, which leads us to weight decomposition. However, previous studies on weight decomposition for LLMs failed to match the performance with practical methods like quantization (Hsu et al., 2022; Yuan et al., 2023; Wang et al., 2024). To tackle this challenge, we propose a novel training-free, decomposition-based weight compression approach called **BitStack**, where we decompose the original weight matrices and iteratively decompose the residuals from the previous approximation. In the decomposition process, we account for the unequal importance of weights (stemming from the high variance in activation channel magnitudes) by scaling the weights before decomposition. We then iteratively apply singular value decomposition (SVD) to decompose the magnitude of the matrices (or residuals) into vectors while preserving their sign matrix, yielding an approximately 1 bit of memory per parameter *residual block* in each iteration. Subsequently, the residual blocks for different weights across various layers are universally sorted and stacked based on their importance to overall performance at the current memory level, stored as basic transmission units in storage. Weight matrices are also treated as stacks, progressively approaching the original matrices as more blocks are added. In this way, BitStack enables a memory-performance trade-off for LLMs by dynamically loading or offloading residual blocks between running memory and storage devices, making LLM deployment feasible in variable memory environments. We conduct extensive evaluations on BitStack across a wide range of tasks, demonstrating that, despite its capability to deploy in variable memory environments, BitStack consistently matches or surpasses the performance of widely adopted compression methods like GPTQ (Frantar et al., 2022) and AWQ (Lin et al., 2024), especially at extreme compression ratios(Figure 1b). To the best of

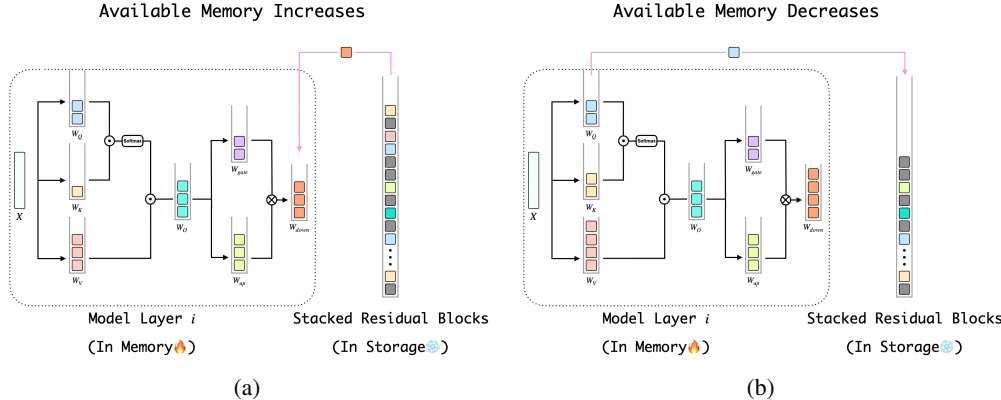

Figure 2: Overview of BitStack. BitStack dynamically loads and offloads *residual blocks* (Figure 3) between RAM and storage devices based on current memory availability. We can load more weight residuals from storage when available memory increases (a), or offload them otherwise (b). The residual blocks for all weights across all layers are universally stored in the same stack on the storage device (grey blocks denote residual blocks for weights in other layers). Note that we omit positional embeddings, normalization layers, and residual connections in the figure for clarity.

our knowledge, BitStack is the first decomposition-based method that closes the performance gap between decomposition-based methods and quantization-based methods.

Our contributions can be summarized as follows: (1) We identify the challenge of deploying LLMs in variable memory environments, which existing model compression methods can not handle. (2) We propose BitStack, a training-free decomposition-based method for model compression that enables megabyte-level memory-performance trade-off for modern LLMs. (3) We conduct extensive experiments on Llama 2, Llama 3, and Llama 3.1 models, ranging in size from 7/8B to 70B, demonstrating that BitStack matches or surpasses the performance of practical quantization-based baselines, particularly at extreme compression ratios.

## 2 BITSTACK

An overview of BitStack is illustrated in Figure 2. BitStack is able to dynamically adjust the size of each weight matrix based on the available memory capacity at the time. When more RAM is freed by other applications, we can retrieve additional residual blocks from a pre-sorted stack and load them into RAM. Conversely, when memory becomes limited, we can offload residual blocks from the model weights (also stored as stacks) back to storage devices in reverse order, ensuring the system remains functional. In the following subsections, we first introduce the decomposition procedure for each weight (or residual) matrix in Section 2.1, and then explain how we sort the residual blocks to be pushed into the universal stack in Section 2.2. A comprehensive overview of BitStack is provided in Algorithm 1.

### 2.1 DECOMPOSING WEIGHTS IN LLMS

In weight decomposition, the objective is to break down weight matrices into sub-matrices to reduce the total number of parameters, with the ability to reconstruct the full matrices during inference. Singular value decomposition (SVD), in particular, is a widely-used and effective method for matrix decomposition due to its ability to capture the most significant components of the weight matrices. Formally, let $\boldsymbol{W} \in \mathbb{R}^{m \times n}$ be a weight matrix in a linear layer, we can decompose $\boldsymbol{W}$ via SVD by:

$$\boldsymbol{W} = \boldsymbol{U}\boldsymbol{\Sigma}\boldsymbol{V}^\top = \sum_{i=1}^{d} \sigma_i \boldsymbol{u}_i \boldsymbol{v}_i^\top \tag{1}$$

where $d = min\{m, n\}$, $\sigma_1 \geq \sigma_2 \geq \cdots \geq \sigma_d$ are the singular values of $\boldsymbol{W}$, and $\boldsymbol{u}_i$ and $\boldsymbol{v}_i$ are the corresponding left and right singular vectors, respectively. We then obtain a rank-$k$ approximation

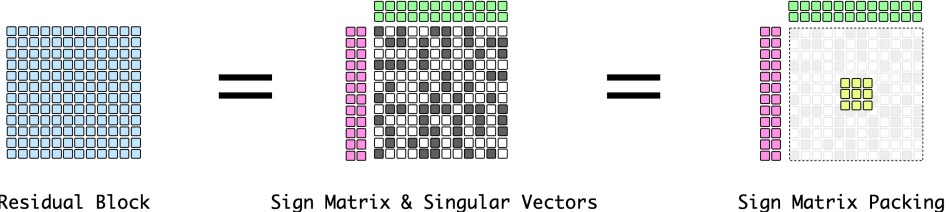

Residual Block       Sign Matrix & Singular Vectors       Sign Matrix Packing

Figure 3: Illustration of a residual block in BitStack. A residual block consists of a sign matrix and singular vectors obtained through absolute value decomposition. The sign matrix can be packed into GPU-supported data types to minimize memory usage. ▦ denotes the sign matrix while ▦ denotes the packed sign matrix.

of $\boldsymbol{W}$:

$$\boldsymbol{W}_{svd} = \boldsymbol{U}_k\boldsymbol{\Sigma}_k\boldsymbol{V}_k^\top = \sum_{i=1}^{k}\sigma_i\boldsymbol{u}_i\boldsymbol{v}_i^\top = \sum_{i=1}^{k}(\sqrt{\sigma_i}\boldsymbol{u}_i)(\sqrt{\sigma_i}\boldsymbol{v}_i^\top) = \boldsymbol{A}\boldsymbol{B}^\top \tag{2}$$

where $\boldsymbol{A} = [\sqrt{\sigma_1}\boldsymbol{u}_1, \ldots, \sqrt{\sigma_k}\boldsymbol{u}_k]$ and $\boldsymbol{B} = [\sqrt{\sigma_1}\boldsymbol{v}_1, \ldots, \sqrt{\sigma_k}\boldsymbol{v}_k]$.

### 2.1.1 ACTIVATION-AWARE DECOMPOSITION

Large language models are known to exhibit outliers in their activations, i.e., the channel variance in $\boldsymbol{X}$ can be high, leading to outputs dominated by these outliers. Fortunately, prior research (Dettmers et al., 2022) has demonstrated that these outliers are often systematically distributed across the activation channels, underscoring the importance of accurately restoring the corresponding weight rows. Lin et al. first proposed scaling the weight matrix using a row-wise scaling vector $\boldsymbol{s}$, which is precomputed with a calibration set to reduce the quantization error of salient weights. Yuan et al. further adopted this method, scaling the weights before applying SVD. In BitStack, we also adopt this methodology to preserve the restoration accuracy of the salient weights. To simplify the process, we do not incorporate any additional searching procedures or hyperparameters to obtain the scaling factors as in previous studies (Lin et al., 2024; Yuan et al., 2023); instead, we compute the scaling factor using the channel-wise $l_2$ norm of $\boldsymbol{X}$. Formally, let $\boldsymbol{X} \in \mathbb{R}^{p \times m}$ represent the input activations for a linear layer, computed using a calibration set, and $\boldsymbol{W} \in \mathbb{R}^{m \times n}$ be the corresponding weight matrix, we compute the scaling factor as follows:

$$\boldsymbol{s} = [\|\boldsymbol{x}_1\|_2, \|\boldsymbol{x}_2\|_2, \cdots, \|\boldsymbol{x}_n\|_2] \tag{3}$$

The inference computation can then be transformed to:

$$\boldsymbol{X}\boldsymbol{W} = \boldsymbol{X}\mathrm{diag}(1/\boldsymbol{s})\mathrm{diag}(\boldsymbol{s})\boldsymbol{W} = \boldsymbol{X}\mathrm{diag}(1/\boldsymbol{s})\boldsymbol{W}_{scaled} \tag{4}$$

And we use $\boldsymbol{W}_{scaled}$ for the subsequent decomposition.

### 2.1.2 ITERATIVE ABSOLUTE VALUE DECOMPOSITION

To reduce the approximation error in each decomposition process, we propose to use absolute value decomposition. In this approach, we first decompose each (scaled) weight matrix into its sign matrix and absolute value matrix; for a weight matrix $\boldsymbol{W} \in \mathbb{R}^{m \times n}$ this is expressed as $\boldsymbol{W} = \boldsymbol{W}_{sign} \odot |\boldsymbol{W}|$. We then apply SVD on $|\boldsymbol{W}|$ while retaining $\boldsymbol{W}_{sign}$. This method enables us to store more information than directly applying SVD on $\boldsymbol{W}$, since we save an additional matrix $\boldsymbol{W}_{sign}$ which is typically large in LLMs. Since $\boldsymbol{W}_{sign}$ consists solely $\pm 1$'s, we can pack $\boldsymbol{W}_{sign}$ to GPU-supported data types for storage and unpack it for use during inference computation. We store the singular vectors in FP16, resulting in an overall memory occupation of approximately 1 bit per parameter when $k \ll min\{m, n\}$ in each decomposition process. A similar technique was employed in recent quantization-aware training research to initialize the weights for 1-bit LLM training (Xu et al., 2024).

Formally, for matrix $\boldsymbol{W} = \boldsymbol{W}_{sign} \odot |\boldsymbol{W}|$, the approximation of $\boldsymbol{W}$ after absolute value decomposition would be:

$$\boldsymbol{W}_{avd} = \boldsymbol{W}_{sign} \odot |\boldsymbol{W}|_{svd} = \boldsymbol{W}_{sign} \odot (\boldsymbol{A}'\boldsymbol{B}'^\top) \tag{5}$$

where $|\boldsymbol{W}|_{svd} = \boldsymbol{U}_k'\boldsymbol{\Sigma}_k'\boldsymbol{V}_k'^\top$, $\boldsymbol{A}' = [\sqrt{\sigma_1'}\boldsymbol{u}_1', \ldots, \sqrt{\sigma_k'}\boldsymbol{u}_k']$ and $\boldsymbol{B}' = [\sqrt{\sigma_1'}\boldsymbol{v}_1', \ldots, \sqrt{\sigma_k'}\boldsymbol{v}_k']$.

To better restore the original matrix $\boldsymbol{W}$, we decompose $\boldsymbol{W}$ over $n$ iterations, progressively decomposing the residuals from the previous approximations. For the $i$-th iteration, we compute:

$$\Delta\boldsymbol{W}^{(i)} = \boldsymbol{W} - \sum_{j=0}^{i-1}\boldsymbol{W}_{iavd}^{(j)} \tag{6}$$

$$\boldsymbol{W}_{iavd}^{(i)} = \Delta\boldsymbol{W}_{avd}^{(i)} = \boldsymbol{W}_{sign}^{(i)} \odot (\boldsymbol{A}'_{(i)}\boldsymbol{B}'^{\top}_{(i)}) \tag{7}$$

where $\boldsymbol{W}_{iavd}^{(0)} = \boldsymbol{0}$. Hence, the overall approximation of $\boldsymbol{W}$ after $n$ iterations is:

$$\boldsymbol{W}_{iavd} = \sum_{i=1}^{n}\boldsymbol{W}_{iavd}^{(i)} = \sum_{i=1}^{n}\boldsymbol{W}_{sign}^{(i)} \odot (\boldsymbol{A}'_{(i)}\boldsymbol{B}'^{\top}_{(i)}) \tag{8}$$

Generally, this approach ensures that the recovery of the original matrix is forward-compatible, allowing us to dynamically load or offload $\boldsymbol{W}_{iavd}^{(i)}$ (termed as *residual blocks* in this paper, illustrated in Figure 3) based on the currently available memory budget, rather than reloading an entirely new model. Additionally, it enables precise size control of the model, as each residual block typically occupies less than a few megabytes, depending on the size of the corresponding weight matrix. See details in Section A.4.

## 2.2 Sorting residual blocks

Having universally decomposed each weight matrix in every layer, it is essential to determine the order in which these residual blocks are loaded from storage into memory to optimize model performance within a given memory budget. To this end, we utilize a small calibration set to calculate perplexity, assessing how much each residual block influences the overall performance. However, solving this sorting problem remains non-trivial, even with this comparison criterion, since the search space is large. For instance, in a model with $L$ layers, each containing $M$ linear layers, and with each weight matrix decomposed over $n$ iterations, there are $n^{LM}$ possible combinations of settings across the various linear layers.

To reduce the search space, we constrain the difference in the number of residual blocks across all weight stacks to no more than 1. This approach facilitates a smooth memory-performance trade-off and promotes effective load balancing when the model is distributed across multiple devices, resulting in a significant reduction of the search space to $nLM$. More specifically, no stack loads the $i + 1$th block until all stacks have loaded the $i$th block. We then sort the relative order of all the $i + 1$th blocks based on their importance, which is measured by the perplexity score after loading this single residual block while keeping all other $i + 1$th blocks for other stacks unloaded. The residual blocks are then placed into a universal stack, ensuring: 1) for all $i$th blocks, blocks with lower measured perplexity scores are on top of those with higher scores; 2) all $i$th blocks are on top of any $i + 1$th ones. This allows a relatively more important block to be loaded when additional memory becomes available. We provide the pseudocode of the sorting process from Line 25 to Line 44 in Algorithm 1.

## 3 Experiments

### 3.1 Evaluation on base models

#### 3.1.1 Settings

**Baselines.** Since our method is training-free, we compare it with two other strong, widely adopted training-free model compression baselines: GPTQ (Frantar et al., 2022) and AWQ (Lin et al., 2024), both of which also require only a small calibration set as in our approach. Note that we do not include the comparison to other decomposition-based methods in the main paper, as they suffer from severe performance degradation under high compression ratios ($1 - \frac{\text{compressed model memory}}{\text{original model memory}}$), and their reported highest compression ratios are significantly lower than those in our study. For example, the highest compression ratios are 30%, 25%, and 60% for FWSVD (Hsu et al., 2022), ASVD (Yuan et al., 2023), and SVD-LLM (Wang et al., 2024), respectively. Furthermore, for the state-of-the-art decomposition-based method, SVD-LLM, the perplexity score increased by 745% and the average performance on zero-shot tasks dropped by 40% at a compression ratio of 60% compared to the orig-

inal model, which falls short of matching the performance of the quantization baselines (Wang et al., 2024). Therefore, we leave the comparison to other decomposition-based methods in Appendix A.6.

Table 1: **Evaluation results of Llama 3.1 8B/70B models.** Perplexity scores on WikiText2 test set and accuracy scores on 6 zero-shot reasoning tasks. (↑): higher is better; (↓): lower is better. We denote the overall compression ratio $(1 - \frac{\text{compressed model memory}}{\text{original model memory}})$ after memory consumption.

| Model | Memory (MB) | Method | Wiki2 (↓) | ARC-e (↑) | ARC-c (↑) | PIQA (↑) | HellaS. (↑) | WinoG. (↑) | LAMBADA (↑) | Avg. (↑) |
|---|---|---|---|---|---|---|---|---|---|---|
| 8B | 15316 | FP 16 | 6.24 | $81.1_{\pm0.8}$ | $53.6_{\pm1.5}$ | $81.2_{\pm0.9}$ | $78.9_{\pm0.4}$ | $73.9_{\pm1.2}$ | $75.8_{\pm0.6}$ | $74.1_{\pm0.9}$ |
| | $3674_{(76\%)}$ | $\text{GPTQ}_{w2}$ | 1.2e6 | $26.0_{\pm0.9}$ | $27.1_{\pm1.3}$ | $51.7_{\pm1.2}$ | $26.0_{\pm0.4}$ | $48.5_{\pm1.4}$ | $0.0_{\pm0.0}$ | $29.9_{\pm0.9}$ |
| | | $\text{AWQ}_{w2}$ | 1.1e6 | $24.9_{\pm0.9}$ | $23.6_{\pm1.2}$ | $49.6_{\pm1.2}$ | $26.2_{\pm0.4}$ | $52.2_{\pm1.4}$ | $0.0_{\pm0.0}$ | $29.4_{\pm0.9}$ |
| | | BitStack | 3.3e3 | $29.3_{\pm0.9}$ | $23.4_{\pm1.2}$ | $53.4_{\pm1.2}$ | $27.9_{\pm0.4}$ | $50.7_{\pm1.4}$ | $0.2_{\pm0.1}$ | $30.8_{\pm0.9}$ |
| | $3877_{(75\%)}$ | $\text{GPTQ}_{w2g128}$ | 1.7e5 | $25.9_{\pm0.9}$ | $26.0_{\pm1.3}$ | $53.9_{\pm1.2}$ | $26.5_{\pm0.4}$ | $49.6_{\pm1.4}$ | $0.0_{\pm0.0}$ | $30.3_{\pm0.9}$ |
| | | $\text{AWQ}_{w2g128}$ | 1.5e6 | $24.6_{\pm0.9}$ | $24.7_{\pm1.3}$ | $50.0_{\pm1.2}$ | $26.4_{\pm0.4}$ | $46.7_{\pm1.4}$ | $0.0_{\pm0.0}$ | $28.7_{\pm0.9}$ |
| | | BitStack | 79.28 | $48.4_{\pm1.0}$ | $26.0_{\pm1.3}$ | $66.5_{\pm1.1}$ | $41.0_{\pm0.5}$ | $57.1_{\pm1.4}$ | $15.5_{\pm0.5}$ | $42.4_{\pm1.0}$ |
| | $4506_{(71\%)}$ | $\text{GPTQ}_{w3}$ | 260.86 | $34.7_{\pm1.0}$ | $24.5_{\pm1.3}$ | $57.6_{\pm1.2}$ | $30.4_{\pm0.5}$ | $53.0_{\pm1.4}$ | $3.0_{\pm0.2}$ | $33.9_{\pm0.9}$ |
| | | $\text{AWQ}_{w3}$ | 17.01 | $67.0_{\pm1.0}$ | $42.9_{\pm1.4}$ | $72.6_{\pm1.0}$ | $67.3_{\pm0.5}$ | $62.6_{\pm1.4}$ | $53.3_{\pm0.7}$ | $61.0_{\pm1.0}$ |
| | | BitStack | 12.55 | $68.5_{\pm1.0}$ | $39.4_{\pm1.4}$ | $75.5_{\pm1.0}$ | $63.4_{\pm0.5}$ | $65.8_{\pm1.3}$ | $66.2_{\pm0.7}$ | $63.1_{\pm1.0}$ |
| | $4709_{(69\%)}$ | $\text{GPTQ}_{w3g128}$ | 38.28 | $55.3_{\pm1.0}$ | $33.9_{\pm1.4}$ | $66.9_{\pm1.1}$ | $53.1_{\pm0.5}$ | $61.9_{\pm1.4}$ | $46.9_{\pm0.7}$ | $53.0_{\pm1.0}$ |
| | | $\text{AWQ}_{w3g128}$ | 8.06 | $74.5_{\pm0.9}$ | $48.4_{\pm1.5}$ | $77.7_{\pm1.0}$ | $73.9_{\pm0.4}$ | $70.6_{\pm1.3}$ | $67.8_{\pm0.7}$ | $68.8_{\pm0.9}$ |
| | | BitStack | 10.91 | $72.7_{\pm0.9}$ | $41.6_{\pm1.4}$ | $76.7_{\pm1.0}$ | $65.9_{\pm0.5}$ | $67.8_{\pm1.3}$ | $69.6_{\pm0.6}$ | $65.7_{\pm1.0}$ |
| | $5338_{(65\%)}$ | $\text{GPTQ}_{w4}$ | 20.88 | $74.7_{\pm0.9}$ | $45.6_{\pm1.5}$ | $77.2_{\pm1.0}$ | $54.6_{\pm0.5}$ | $64.5_{\pm1.3}$ | $40.9_{\pm0.7}$ | $59.6_{\pm1.0}$ |
| | | $\text{AWQ}_{w4}$ | 7.12 | $78.4_{\pm0.8}$ | $51.1_{\pm1.5}$ | $79.9_{\pm0.9}$ | $77.5_{\pm0.4}$ | $73.3_{\pm1.2}$ | $70.6_{\pm0.6}$ | $71.8_{\pm0.9}$ |
| | | BitStack | 8.39 | $76.6_{\pm0.9}$ | $47.9_{\pm1.5}$ | $79.0_{\pm1.0}$ | $71.6_{\pm0.4}$ | $69.6_{\pm1.3}$ | $76.1_{\pm0.6}$ | $70.1_{\pm0.9}$ |
| | $5541_{(64\%)}$ | $\text{GPTQ}_{w4g128}$ | 6.83 | $78.6_{\pm0.8}$ | $51.5_{\pm1.5}$ | $79.1_{\pm0.9}$ | $77.0_{\pm0.4}$ | $71.2_{\pm1.3}$ | $72.9_{\pm0.6}$ | $71.7_{\pm0.9}$ |
| | | $\text{AWQ}_{w4g128}$ | 6.63 | $79.3_{\pm0.8}$ | $51.2_{\pm1.5}$ | $81.0_{\pm0.9}$ | $78.2_{\pm0.4}$ | $72.1_{\pm1.3}$ | $74.2_{\pm0.6}$ | $72.7_{\pm0.9}$ |
| | | BitStack | 8.14 | $77.6_{\pm0.9}$ | $49.7_{\pm1.5}$ | $79.5_{\pm0.9}$ | $72.4_{\pm0.4}$ | $70.6_{\pm1.3}$ | $76.0_{\pm0.6}$ | $71.0_{\pm0.9}$ |
| 70B | 134570 | FP 16 | 2.81 | $86.7_{\pm0.7}$ | $64.8_{\pm1.4}$ | $84.3_{\pm0.8}$ | $85.1_{\pm0.4}$ | $79.8_{\pm1.1}$ | $79.2_{\pm0.6}$ | $80.0_{\pm0.8}$ |
| | $20356_{(85\%)}$ | $\text{GPTQ}_{w2}$ | NaN | $24.8_{\pm0.9}$ | $26.2_{\pm1.3}$ | $50.8_{\pm1.2}$ | $26.4_{\pm0.4}$ | $51.4_{\pm1.4}$ | $0.0_{\pm0.0}$ | $29.9_{\pm0.9}$ |
| | | $\text{AWQ}_{w2}$ | 9.6e5 | $25.0_{\pm0.9}$ | $25.5_{\pm1.3}$ | $51.7_{\pm1.2}$ | $26.6_{\pm0.4}$ | $50.4_{\pm1.4}$ | $0.0_{\pm0.0}$ | $29.9_{\pm0.9}$ |
| | | BitStack | 1.0e3 | $27.9_{\pm0.9}$ | $23.9_{\pm1.2}$ | $52.3_{\pm1.2}$ | $30.4_{\pm0.5}$ | $49.6_{\pm1.4}$ | $2.6_{\pm0.2}$ | $31.1_{\pm0.9}$ |
| | $22531_{(83\%)}$ | $\text{GPTQ}_{w2g128}$ | 4.4e5 | $23.9_{\pm0.9}$ | $25.6_{\pm1.3}$ | $51.1_{\pm1.2}$ | $26.4_{\pm0.4}$ | $50.4_{\pm1.4}$ | $0.0_{\pm0.0}$ | $29.6_{\pm0.9}$ |
| | | $\text{AWQ}_{w2g128}$ | 1.8e6 | $24.9_{\pm0.9}$ | $26.2_{\pm1.3}$ | $51.3_{\pm1.2}$ | $26.8_{\pm0.4}$ | $49.4_{\pm1.4}$ | $0.0_{\pm0.0}$ | $29.8_{\pm0.9}$ |
| | | BitStack | 8.50 | $76.8_{\pm0.9}$ | $50.6_{\pm1.5}$ | $77.9_{\pm1.0}$ | $74.2_{\pm0.4}$ | $73.7_{\pm1.2}$ | $73.2_{\pm0.6}$ | $71.1_{\pm0.9}$ |
| | $28516_{(79\%)}$ | $\text{GPTQ}_{w3}$ | 3.7e6 | $24.7_{\pm0.9}$ | $26.8_{\pm1.3}$ | $51.1_{\pm1.2}$ | $26.3_{\pm0.4}$ | $50.5_{\pm1.4}$ | $0.0_{\pm0.0}$ | $29.9_{\pm0.9}$ |
| | | $\text{AWQ}_{w3}$ | 10.76 | $57.4_{\pm1.0}$ | $37.0_{\pm1.4}$ | $71.1_{\pm1.1}$ | $63.8_{\pm0.5}$ | $59.0_{\pm1.4}$ | $49.5_{\pm0.7}$ | $56.3_{\pm1.0}$ |
| | | BitStack | 6.38 | $81.7_{\pm0.8}$ | $56.7_{\pm1.4}$ | $81.8_{\pm0.9}$ | $79.3_{\pm0.4}$ | $76.6_{\pm1.2}$ | $76.8_{\pm0.6}$ | $75.5_{\pm0.9}$ |
| | $30691_{(77\%)}$ | $\text{GPTQ}_{w3g128}$ | 4.4e5 | $24.2_{\pm0.9}$ | $24.2_{\pm1.3}$ | $51.7_{\pm1.2}$ | $26.0_{\pm0.4}$ | $49.3_{\pm1.4}$ | $0.0_{\pm0.0}$ | $29.2_{\pm0.9}$ |
| | | $\text{AWQ}_{w3g128}$ | 4.68 | $84.0_{\pm0.8}$ | $60.6_{\pm1.4}$ | $83.1_{\pm0.9}$ | $82.5_{\pm0.4}$ | $79.2_{\pm1.1}$ | $75.8_{\pm0.6}$ | $77.5_{\pm0.9}$ |
| | | BitStack | 5.94 | $82.6_{\pm0.8}$ | $58.3_{\pm1.4}$ | $82.9_{\pm0.9}$ | $80.9_{\pm0.4}$ | $78.8_{\pm1.1}$ | $78.4_{\pm0.6}$ | $77.0_{\pm0.9}$ |
| | $36676_{(73\%)}$ | $\text{GPTQ}_{w4}$ | NaN | $24.9_{\pm0.9}$ | $25.3_{\pm1.3}$ | $51.4_{\pm1.2}$ | $26.8_{\pm0.4}$ | $51.1_{\pm1.4}$ | $0.0_{\pm0.0}$ | $29.9_{\pm0.9}$ |
| | | $\text{AWQ}_{w4}$ | 4.24 | $83.4_{\pm0.8}$ | $61.3_{\pm1.4}$ | $83.5_{\pm0.9}$ | $83.4_{\pm0.4}$ | $63.5_{\pm1.4}$ | $69.1_{\pm0.6}$ | $74.0_{\pm0.9}$ |
| | | BitStack | 4.97 | $84.8_{\pm0.7}$ | $61.4_{\pm1.4}$ | $83.2_{\pm0.9}$ | $82.1_{\pm0.4}$ | $79.3_{\pm1.1}$ | $79.4_{\pm0.6}$ | $78.4_{\pm0.9}$ |
| | $38851_{(71\%)}$ | $\text{GPTQ}_{w4g128}$ | 6.5e4 | $23.4_{\pm0.9}$ | $27.3_{\pm1.3}$ | $51.9_{\pm1.2}$ | $26.6_{\pm0.4}$ | $49.9_{\pm1.4}$ | $0.0_{\pm0.0}$ | $29.8_{\pm0.9}$ |
| | | $\text{AWQ}_{w4g128}$ | 3.27 | $86.6_{\pm0.7}$ | $63.3_{\pm1.4}$ | $83.9_{\pm0.9}$ | $84.4_{\pm0.4}$ | $78.8_{\pm1.1}$ | $77.3_{\pm0.6}$ | $79.1_{\pm0.8}$ |
| | | BitStack | 4.96 | $85.1_{\pm0.7}$ | $61.3_{\pm1.4}$ | $83.5_{\pm0.9}$ | $82.6_{\pm0.4}$ | $78.8_{\pm1.1}$ | $78.7_{\pm0.6}$ | $78.3_{\pm0.9}$ |

**Evaluation.** We evaluate our approach alongside the baselines on the well-known Llama2, Llama3, and Llama3.1 series (Touvron et al., 2023; Dubey et al., 2024), with model sizes ranging from 7/8B to 70B parameters. We conduct the evaluations by computing the perplexity score on the WikiText2 testset (Merity et al., 2016), and accuracy scores on a range of zero-shot reasoning tasks including PIQA (Bisk et al., 2020), HellaSwag (Zellers et al., 2019), WinoGrande (Sakaguchi et al., 2021), ARC-easy and ARC-challenge (Clark et al., 2018), LAMBADA(OpenAI) (Radford et al., 2019). The zero-shot tasks are evaluated using the LM Evaluation Harness (Gao et al., 2024) with default parameters.

**Implementation details** For both our approach and the baselines, we randomly sample 256 samples with sequence length 2048 from the WikiText2 (Merity et al., 2016) training set to serve as the calibration set. The baselines are implemented using zeropoint quantization, and we report their evaluation results both with and without group quantization to ensure optimal performance and fair comparison. For group quantization, we use a group size of 128, which provides a significant performance boost to the baselines. It is worth noting that the performance of GPTQ on Llama 3 and Llama 3.1 models can be unstable and may occasionally collapse. This variability is likely due to the Llama 3 series being more sensitive to compression (Huang et al., 2024b), necessitating a larger number of calibration samples in GPTQ for optimal performance. By default, in BitStack, we per-

form iterative absolute value decomposition (IAVD) on each weight matrix over 16 iterations, saving the sign matrix and the first 16 singular vectors in each decomposition process. Additionally, for sorting the residual blocks in BitStack, we sample a smaller set of 32 samples from the WikiText2 training set to evaluate the importance of each residual block. All experiments are conducted on a node with 8 NVIDIA H800 GPUs.

### 3.1.2 RESULTS

Evaluation results of both the perplexity scores and zero-shot performance of Llama 3.1/Llama 2/Llama 3 models are presented in Table 1, 2 and 3, respectively. Since BitStack allows for megabyte-level size control, we align the model sizes of the BitStack-compressed models with those of the different baselines for a fair comparison. Specifically, we utilize the largest size that does not exceed the baselines' sizes.

**BitStack performs better at extreme compression ratios.** As shown in the tables, BitStack delivers superior or comparable performance with strong quantization baselines across different compression ratios, despite having the advantage that it only needs to compress and store once and can dynamically adjust its memory consumption at a megabyte level. More specifically, BitStack models constantly outperform the baselines at extremely high compression ratios. For 7/8B models, BitStack constantly outperforms GPTQ models below 4-bit-level and AWQ models below 3-bit-level. For 7/8B models, BitStack outperforms the best 2-bit baselines with group quantization by an absolute margin of 12.1(Llama 3.1), 22.3(Llama 2), 10.4(Llama 3) on average performance in zero-shot tasks. This advantage is even more pronounced in larger models; for example, on the Llama 3.1 70B, BitStack retains 89%of the performance of the original FP16 models, surpassing the best baseline by a substantial margin of 41.3 on zero-shot tasks.

**BitStack maintains strong performance at lower compression ratios.** While quantization baselines excel at lower compression ratios, BitStack maintains comparable effectiveness, even with group quantization, which significantly enhances the performance of these quantization methods. For instance, at the lowest compression ratio (64%) in our experiments, BitStack Llama 3.1 8B and 70B models can recover 96% and 98% of the zero-shot performance of the original FP16 model, respectively. Although they exhibit slightly higher perplexity scores, they only fall short of the best baselines by a negligible 1.7 and 0.8 absolute average score on zero-shot tasks. As shown in the tables, the gap consistently narrows as the model size increases. For instance, when compared to the best baseline with group quantization, the gap in zero-shot tasks decreases from 1.7 to 1.1 to 0.2 for Llama 2 models with 7B, 13B, and 70B parameters, respectively(Table. 2). It can be seen that BitStack demonstrates particularly strong performance with larger models. For 70B models, it consistently outperforms the baselines without group quantization across all compression ratios in our experiments.

## 3.2 EVALUATION ON INSTRUCTION-TUNED MODELS

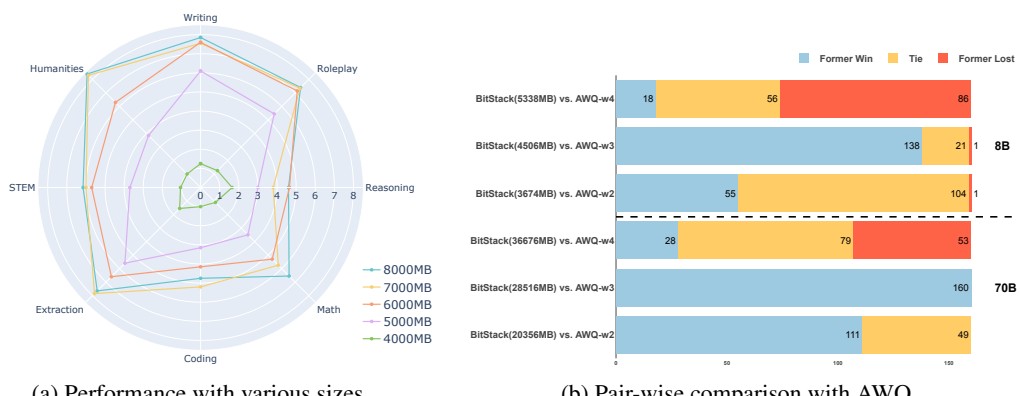

(a) Performance with various sizes.   (b) Pair-wise comparison with AWQ.

Figure 4: Evaluation results of BitStack Llama 3.1 Instruct 8B/70B models on MT-Bench, assessed by `gpt-4o`. (a) demonstrates the single-answer grading results across various sizes of the 8B model loaded by BitStack, while (b) illustrates the pairwise comparison results against AWQ at different compression ratios for both the 8B and 70B models.

### 3.2.1 SETTINGS

To assess the generalization capability of our method, we conduct further experiments on instruction-tuned models. Specifically, we apply compression to the Llama 3.1 Instruct 8B and 70B models using both our approach and AWQ, which has been shown to be a stronger baseline in the previous section. We follow the procedure in Zheng et al. (2023) and evaluate the compressed models on MT-Bench (Zheng et al., 2023), which consists of 80 multi-turn common user prompts, covering writing, roleplay, extraction, reasoning, math, coding, knowledge I (STEM), and knowledge II (humanities/social science). We use OpenAI `gpt-4o` as the judging model to evaluate the model answers.

### 3.2.2 RESULTS

Figure 4a illustrates the evaluation results on BitStack compressed Llama-3.1-Instruct-8B model with {4000, 5000, 6000, 7000, 8000} megabytes. The results show a clear trend across all domains: increasing the model size (by loading more residual blocks from storage) consistently improves performance. This underscores that while BitStack facilitates fine-grained memory-performance trade-offs, the performance improvement spans all domains comprehensively. When compared to AWQ, BitStack demonstrates a similar trend at various compression ratios as seen with the base models. As shown in Figure 4b, at extremely high compression ratios—approximately at the 2-bit level—BitStack models can occasionally generate reasonable answers, whereas the AWQ compressed model fails to produce coherent text. This distinction becomes even more pronounced at the 3-bit level, where the BitStack model consistently generates high-quality responses, while the AWQ model still outputs gibberish. At lower compression ratios (4-bit level), where quantization-based methods excel, BitStack outperforms or matches the baseline on about $\frac{1}{2}$ of the samples for the 8B model and about $\frac{2}{3}$ for the 70B model. We provide extra qualitative results in Section A.3.

### 3.3 ABLATION STUDY AND ANALYSIS

In this section, we conduct an ablation study to evaluate the impact of each component within our proposed approach. We assess performance by plotting perplexity and average zero-shot accuracy curves to measure the model's effectiveness at different memory footprints. For these experiments, we use the BitStack Llama 3.1 8B model and evaluate performance with a memory stride of 500MB. Additionally, we provide further discussion on the minimal transmission units in BitStack in Section A.4 and analyze the inference throughput of BitStack models in Section A.5.

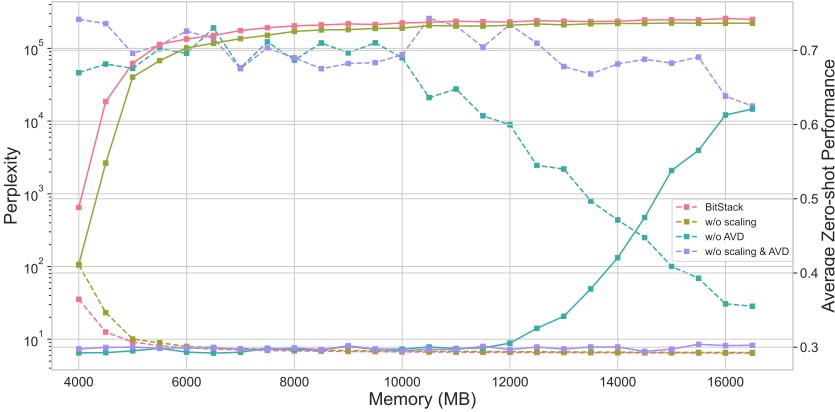

Figure 5: Perplexity and average zero-shot performance of BitStack Llama 3.1 8B with or without activation-aware scaling and absolute value decomposition(AVD). In the "w/o scaling" experiments, no scaling is applied as in Eq. 4; in the "w/o AVD" experiments, vanilla SVD is used instead of AVD as in Eq. 5. For vanilla SVD, we set $k' = k + \frac{m \times n}{16 \times (m+n)}$ (for $W \in \mathbb{R}^{m \times n}$) to ensure the size of each residual block matches that of the main experiments. Solid lines represent average zero-shot performance, while dotted lines represent perplexity scores.

**Impact of each component.** As shown in Figure 5, activation-aware scaling consistently improves model performance across all compression ratios, with particularly strong effects at higher compression ratios. For instance, it leads to an 8-point improvement in average zero-shot performance at a

memory footprint of 4000MB. Regarding absolute value decomposition (AVD), we ablate it by replacing it with vanilla SVD, using a larger number of kept singular vectors $k$ to match the sizes of residual blocks. The figure shows that when AVD is replaced with SVD, the model performance degrades significantly, collapsing until a memory footprint of 12,000MB, which corresponds to a compression ratio of 22%, even with activation-aware scaling applied. This highlights that AVD significantly enhances approximation accuracy during the decomposition process compared to SVD under the same memory constraints, enabling the model to maintain strong performance at high compression ratios. When both activation-aware scaling and AVD are removed, the model collapses across all compression ratios, underscoring the critical importance of these components. Note that we use the same sorting approach as we proposed in Section 2.2 for all these experiments.

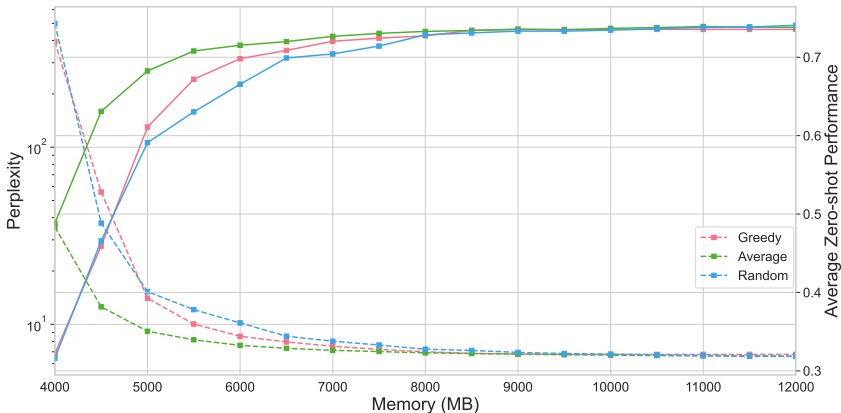

Figure 6: Perplexity and average zero-shot performance of BitStack Llama 3.1 8B with 3 different sorting approaches for residual blocks. Solid lines represent average zero-shot performance, while dotted lines represent perplexity scores.

**Impact of the sorting algorithm for residual blocks.** To reduce the search space for sorting residual blocks, we propose constraining the length difference between weight stacks to no more than 1(as detailed in Section 2.2, referred to as **Average**). We compare this approach to two alternatives: 1) **Random**, which randomly shuffles the universal residual stack without any search process; 2) **Greedy**, which evaluates each weight stack at each level (number of residual blocks) while freezing all other weight stacks at a level of $\frac{n}{2}$, and utilize the current perplexity as the importance score for corresponding stack at that level, which also has a search space of $nLM$. We provide visualization of resulting weight stacks of the three sorting approaches in Section A.7. As shown in Figure. 6, as the memory footprint goes up, all three approaches converge as most residual blocks are loaded into the model. However, at lower memory footprints($< 8000$MB), **Average** significantly outperforms both baselines, surpassing the best baseline by 16, 16, and 7 points in absolute zero-shot performance at 4000MB, 4500MB, and 5000MB, respectively. In addition to excelling at high compression ratios, **Average** also provides better load balancing, as the memory footprint of each block varies minimally, making it easier to deploy in distributed scenarios.

**Ablation on calibration set size $n$.** We compute the scaling vector $s$ using various sizes of the calibration set, as shown in Figure. 7a. The figure demonstrates that BitStack is robust to changes in calibration set size, as the curves align almost perfectly across different sizes, particularly as the memory footprint increases. Interestingly, BitStack even performs slightly better with a smaller calibration set size of 64 in extreme compression scenarios, such as with a memory footprint of 4000MB.

**Ablation on number of kept singular vectors $k$ in each decomposition process.** Generally, a larger $k$ in SVD indicates a better approximation in each decomposition process, but at the cost of increased memory usage, as the singular vectors are stored in FP16. Figure 7b illustrates the performance when setting $k$ to $\{1, 4, 8, 16, 32\}$. As shown in the figure, keeping only the largest singular value and its corresponding vectors is insufficient for a good approximation, leading to performance degradation. On the other hand, increasing $k$ results in fewer residual blocks being loaded at the same memory footprint, limiting model performance. This is evident from the figure,

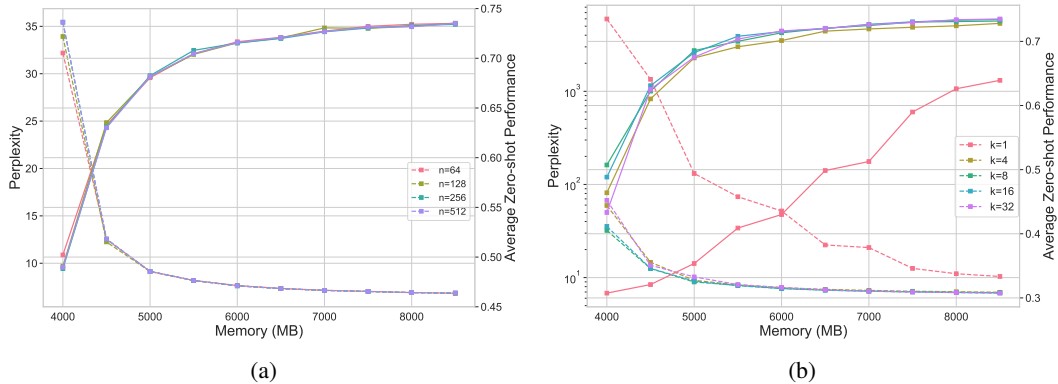

Figure 7: Perplexity and average zero-shot performance of BitStack Llama 3.1 8B with various calibration set sizes $n$ (a) and number of singular vectors $k$ (b). Solid lines represent average zero-shot performance, while dotted lines represent perplexity scores.

as increasing $k$ beyond 1 provides no significant performance improvement. Overall, $k = 16$ strikes a good balance between approximation accuracy and memory consumption.

## 4   RELATED WORK

**Fixed ratio weight compression.**   As discussed in Section 1, we categorize weight compression approaches such as quantization, pruning, and distillation under fixed ratio weight compression. Quantization-based methods compress weights by reducing precision, pruning techniques compress by directly modifying the model structure (e.g., reducing the number of layers or hidden dimensions), and distillation methods involve training a smaller model on the outputs of the original model. The latter two approaches, as well as quantization methods for higher compression ratios, typically require extensive training (Ma et al., 2023; Xia et al., 2023; Muralidharan et al., 2024), which can be computationally expensive when compressing models for multiple compression ratios (Shao et al., 2023; Tseng et al., 2024; Egiazarian et al., 2024; Huang et al., 2024a). Furthermore, models compressed by these methods are poorly suited for variable memory environments due to their fixed memory usage, preventing efficient utilization of available capacity.

**Adaptive ratio weight compression.**   Weight decomposition methods are more suitable for adaptive ratio weight compression due to their forward-compatible nature, as the approximation improves with the inclusion of more singular vectors in SVD. However, current decomposition-based weight compression approaches for LLMs tend to collapse at high compression ratios (Hsu et al., 2022; Yuan et al., 2023; Wang et al., 2024), rendering them impractical for real-world deployment. In this work, we bridge the performance gap between decomposition-based methods and practical quantization-based approaches, making LLM deployment in variable memory environments feasible.

## 5   CONCLUSION

In this paper, we highlight the challenge of deploying compressed large language models in variable memory environments and propose BitStack, a decomposition-based compression approach designed to address this issue. BitStack enables megabyte-level memory-performance trade-offs in a training-free manner, requiring only a small calibration set. Additionally, BitStack is simple to implement, with the decomposition of 70B models being achievable on a single GPU. Despite its flexibility in memory footprint, BitStack consistently matches or surpasses the performance of practical baselines, making it a viable solution for real-world applications. We believe that BitStack represents a new paradigm for LLM deployment on local devices, providing not only efficient memory management but also strong performance within the given memory budget.

ACKNOWLEDGMENTS

This work was supported by the National Key Research and Development Program of China (No. U24B20181).

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

## A APPENDIX

We provide further details of our approach in this appendix as follows:

- Section A.1, the overall algorithm of BitStack.
- Section A.2, evaluation results of Llama 2 and Llama 3 models.
- Section A.3, qualitative results of BitStack Llama 3.1 Instruct 8B and 70B models.
- Section A.4, discussion on minimal transmission units in BitStack.
- Section A.5, analysis of inference throughput of BitStack.
- Section A.6, comparison to other decomposition-based approaches.
- Section A.7, visualizations of weight stack in different sorting approaches.

## A.1 OVERALL ALGORITHM OF BITSTACK

---

**Algorithm 1** BitStack

---

**Input:** A model $\mathcal{M}$ with $L$ layers each consists $M$ weight matrices with weight matrices $\{\boldsymbol{W}^{(11)}, \cdots, \boldsymbol{W}^{(LM)}\}$ and corresponding activations $\{\boldsymbol{X}^{(11)}, \cdots, \boldsymbol{X}^{(LM)}\}$ in a calibration set $\boldsymbol{X}$. Number of decompose iterations $n$, number of singular values $k$ retained in SVD. And extra small calibration set $\boldsymbol{X}'$ for sorting.

**Output:** Sorted residual block stack $S$

 1: **procedure** ACTIVATION-AWARE WEIGHT SCALING
 2:     **for** $l \leftarrow 1$ **to** $L$ **do**
 3:         **for** $m \leftarrow 1$ **to** $M$ **do**
 4:             $\boldsymbol{s}^{(lm)} = \left[ \|\boldsymbol{x}_1^{(lm)}\|_2, \|\boldsymbol{x}_2^{(lm)}\|_2, \cdots, \|\boldsymbol{x}_n^{(li)}\|_2 \right]$         ▷ Eq. (3)
 5:             $\boldsymbol{W}^{(lm)}_{scaled} = \mathrm{diag}(\boldsymbol{s}^{(lm)})\boldsymbol{W}^{(lm)}$         ▷ Eq. (4)
 6:         **end for**
 7:     **end for**
 8: **end procedure**
 9: Now that the weight matrices are scaled, we omit the $_{scaled}$.subscript for clarity.
10: **procedure** ITERATIVE ABSOLUTE DECOMPOSITION
11:     **for** $l \leftarrow 1$ **to** $L$ **do**
12:         **for** $m \leftarrow 1$ **to** $M$ **do**
13:             Initialize a empty weight stack $S^{(lm)}$
14:             **for** $i \leftarrow 1$ **to** $n$ **do**
15:                 $S^{(lm)}.\mathtt{push}\,(W^{(lm)(i)}_{isavd})$         ▷ Eq. (6)
16:             **end for**
17:             Initialize a empty stack $S^{(lm)}_{temp}$ to temporarily store the elements for subsequent sorting.
18:             **for** $i \leftarrow 1$ **to** $n-1$ **do**
19:                 $S^{(lm)}_{temp}.\mathtt{push}\,(S^{(lm)}.\mathtt{pop}\,())$
20:             **end for**
21:         **end for**
22:     **end for**
23: **end procedure**
24: Initialize an empty universal residual block stack $S$.
25: **procedure** RESIDUAL BLOCK SORTING
26:     **for** $i \leftarrow 1$ **to** $n-1$ **do**
27:         **for** $l \leftarrow 1$ **to** $L$ **do**
28:             **for** $m \leftarrow 1$ **to** $M$ **do**
29:                 $S^{(lm)}.\mathtt{push}\,(S^{(lm)}_{temp}.\mathtt{pop}\,())$     ▷ Push a new residual block for assessing.
30:                 $W^{(lm)(i)}_{isavd}.\mathtt{importance} = \mathtt{compute\_perplexity}\,(\mathcal{M},\ \boldsymbol{X}')$ ▷ Measure influence.
31:                 $S^{(lm)}_{temp}.\mathtt{push}\,(S^{(lm)}.\mathtt{pop}\,())$       ▷ Pop before evaluating the next weight.
32:             **end for**
33:         **end for**
34:         $\mathtt{sorted\_ith\_bit\_residual\_blocks} = \mathtt{sort}\,(W^{(i)}_{isavd},\ \mathtt{key=importance})$
35:         **for** $\mathtt{block}$ **in** $\mathtt{sorted\_ith\_bit\_residual\_blocks}$ **do**
36:             $S.\mathtt{push}\,(\mathtt{block})$         ▷ Push important blocks first.
37:         **end for**
38:         **for** $l \leftarrow 1$ **to** $L$ **do**
39:             **for** $m \leftarrow 1$ **to** $M$ **do**
40:                 $S^{(lm)}.\mathtt{push}\,(S^{(lm)}_{temp}.\mathtt{pop}\,())$    ▷ Push all stacks before assessing $(i+1)$th blocks.
41:             **end for**
42:         **end for**
43:     **end for**
44: **end procedure**
45: **return** $S.\mathtt{reverse}\,()$ ▷ Position residual blocks with lower bit levels and higher importance at the top.

---

Algorithm 1 illustrates the pseudocode for the overall algorithm of BitStack. Specifically, Lines 1 to 8 describe the activation-aware weight scaling process, as introduced in Section 2.1.1. This scaling is applied only once before the iterative absolute decomposition. Lines 9 to 23 detail the iterative absolute decomposition process, as explained in Section 2.1.1 where each scaled weight matrix is decomposed into $n$ *residual blocks* and stored as a stack. Finally, Lines 24 to 44 demonstrate the

residual block sorting process, as discussed in Section 2.2, where the influence of each residual block is evaluated while keeping all other weight stacks at the same level. The blocks are then sorted by their evaluated importance and placed into a universal stack.

## A.2 EVALUATIONS OF LLAMA2 AND LLAMA3 MODELS

Table 2: **Evaluation results of Llama 2 7B/13B/70B models.** Perplexity scores on WikiText2 test set and accuracy scores on 6 zero-shot reasoning tasks. (↑): higher is better; (↓): lower is better. We denote the overall compression ratio ($1 - \frac{\text{compressed model memory}}{\text{original model memory}}$) after memory consumption.

| Model | Memory (MB) | Method | Wiki2 (↓) | ARC-e (↑) | ARC-c (↑) | PIQA (↑) | HellaS. (↑) | WinoG. (↑) | LAMBADA (↑) | Avg. (↑) |
|---|---|---|---|---|---|---|---|---|---|---|
| 7B | 12852 | FP 16 | 5.47 | $74.5_{\pm0.9}$ | $46.2_{\pm1.5}$ | $79.1_{\pm0.9}$ | $76.0_{\pm0.4}$ | $69.1_{\pm1.3}$ | $73.9_{\pm0.6}$ | $69.8_{\pm0.9}$ |
| | $2050_{(84\%)}$ | GPTQ$_{w2}$ | 2.8e4 | $26.5_{\pm0.9}$ | $\mathbf{27.6}_{\pm1.3}$ | $48.4_{\pm1.2}$ | $25.9_{\pm0.4}$ | $50.3_{\pm1.4}$ | $0.0_{\pm0.0}$ | $29.8_{\pm0.9}$ |
| | | AWQ$_{w2}$ | 1.8e5 | $26.3_{\pm0.9}$ | $26.7_{\pm1.3}$ | $50.9_{\pm1.2}$ | $26.5_{\pm0.4}$ | $49.3_{\pm1.4}$ | $0.0_{\pm0.0}$ | $30.0_{\pm0.9}$ |
| | | BitStack | **29.93** | $\mathbf{32.3}_{\pm1.0}$ | $25.6_{\pm1.3}$ | $\mathbf{62.4}_{\pm1.1}$ | $\mathbf{42.8}_{\pm0.5}$ | $\mathbf{53.6}_{\pm1.4}$ | $\mathbf{24.7}_{\pm0.6}$ | $\mathbf{40.2}_{\pm1.0}$ |
| | $2238_{(83\%)}$ | GPTQ$_{w2g128}$ | 156.37 | $28.2_{\pm0.9}$ | $27.1_{\pm1.3}$ | $51.7_{\pm1.2}$ | $28.0_{\pm0.4}$ | $51.1_{\pm1.4}$ | $0.3_{\pm0.1}$ | $31.1_{\pm0.9}$ |
| | | AWQ$_{w2g128}$ | 2.3e5 | $25.8_{\pm0.9}$ | $26.7_{\pm1.3}$ | $50.2_{\pm1.2}$ | $26.1_{\pm0.4}$ | $49.8_{\pm1.4}$ | $0.0_{\pm0.0}$ | $29.8_{\pm0.9}$ |
| | | BitStack | **12.49** | $\mathbf{51.8}_{\pm1.0}$ | $\mathbf{30.1}_{\pm1.3}$ | $\mathbf{71.1}_{\pm1.1}$ | $\mathbf{53.0}_{\pm0.5}$ | $\mathbf{61.1}_{\pm1.4}$ | $\mathbf{53.3}_{\pm0.7}$ | $\mathbf{53.4}_{\pm1.0}$ |
| | $2822_{(78\%)}$ | GPTQ$_{w3}$ | 9.38 | $58.1_{\pm1.0}$ | $34.0_{\pm1.4}$ | $71.9_{\pm1.0}$ | $61.7_{\pm0.5}$ | $60.6_{\pm1.4}$ | $53.3_{\pm0.7}$ | $56.6_{\pm1.0}$ |
| | | AWQ$_{w3}$ | 14.33 | $52.7_{\pm1.0}$ | $33.0_{\pm1.4}$ | $68.3_{\pm1.1}$ | $56.3_{\pm0.5}$ | $59.3_{\pm1.4}$ | $36.3_{\pm0.7}$ | $51.0_{\pm1.0}$ |
| | | BitStack | **7.45** | $\mathbf{62.5}_{\pm1.0}$ | $\mathbf{37.5}_{\pm1.4}$ | $\mathbf{74.8}_{\pm1.0}$ | $\mathbf{67.0}_{\pm0.5}$ | $\mathbf{66.5}_{\pm1.3}$ | $\mathbf{68.5}_{\pm0.6}$ | $\mathbf{62.8}_{\pm1.0}$ |
| | $3010_{(77\%)}$ | GPTQ$_{w3g128}$ | 922.54 | $26.3_{\pm0.9}$ | $25.3_{\pm1.3}$ | $52.4_{\pm1.2}$ | $27.4_{\pm0.4}$ | $49.0_{\pm1.4}$ | $0.1_{\pm0.0}$ | $30.1_{\pm0.9}$ |
| | | AWQ$_{w3g128}$ | **6.14** | $\mathbf{70.2}_{\pm1.0}$ | $\mathbf{43.7}_{\pm1.4}$ | $\mathbf{78.0}_{\pm1.0}$ | $\mathbf{73.9}_{\pm0.4}$ | $\mathbf{67.6}_{\pm1.3}$ | $\mathbf{71.4}_{\pm0.6}$ | $\mathbf{67.5}_{\pm1.0}$ |
| | | BitStack | 7.10 | $63.8_{\pm1.0}$ | $38.2_{\pm1.4}$ | $76.0_{\pm1.0}$ | $68.4_{\pm0.5}$ | $65.9_{\pm1.3}$ | $70.7_{\pm0.6}$ | $63.8_{\pm1.0}$ |
| | $3594_{(72\%)}$ | GPTQ$_{w4}$ | 5.91 | $\mathbf{71.8}_{\pm0.9}$ | $43.7_{\pm1.4}$ | $77.7_{\pm1.0}$ | $74.5_{\pm0.4}$ | $68.7_{\pm1.3}$ | $71.1_{\pm0.6}$ | $67.9_{\pm1.0}$ |
| | | AWQ$_{w4}$ | **5.81** | $70.9_{\pm0.9}$ | $\mathbf{44.5}_{\pm1.5}$ | $\mathbf{78.5}_{\pm1.0}$ | $\mathbf{74.8}_{\pm0.4}$ | $69.2_{\pm1.3}$ | $71.5_{\pm0.6}$ | $\mathbf{68.2}_{\pm1.0}$ |
| | | BitStack | 6.36 | $67.0_{\pm1.0}$ | $41.4_{\pm1.4}$ | $77.1_{\pm1.0}$ | $71.4_{\pm0.5}$ | $\mathbf{69.5}_{\pm1.3}$ | $\mathbf{73.1}_{\pm0.6}$ | $66.6_{\pm1.0}$ |
| | $3782_{(71\%)}$ | GPTQ$_{w4g128}$ | 5.73 | $\mathbf{73.6}_{\pm0.9}$ | $\mathbf{45.3}_{\pm1.5}$ | $\mathbf{78.7}_{\pm1.0}$ | $75.4_{\pm0.4}$ | $67.6_{\pm1.3}$ | $72.7_{\pm0.6}$ | $\mathbf{68.9}_{\pm0.9}$ |
| | | AWQ$_{w4g128}$ | **5.61** | $73.3_{\pm0.9}$ | $45.2_{\pm1.5}$ | $78.6_{\pm1.0}$ | $75.2_{\pm0.4}$ | $\mathbf{68.7}_{\pm1.3}$ | $72.7_{\pm0.6}$ | $\mathbf{68.9}_{\pm0.9}$ |
| | | BitStack | 6.27 | $67.8_{\pm1.0}$ | $43.3_{\pm1.4}$ | $77.2_{\pm1.0}$ | $72.2_{\pm0.4}$ | $68.6_{\pm1.3}$ | $\mathbf{73.9}_{\pm0.6}$ | $67.2_{\pm1.0}$ |
| 13B | 24825 | FP 16 | 4.88 | $77.4_{\pm0.9}$ | $49.1_{\pm1.5}$ | $80.5_{\pm0.9}$ | $79.4_{\pm0.4}$ | $72.2_{\pm1.3}$ | $76.8_{\pm0.6}$ | $72.6_{\pm0.9}$ |
| | $3659_{(85\%)}$ | GPTQ$_{w2}$ | 1.2e4 | $26.4_{\pm0.9}$ | $\mathbf{28.2}_{\pm1.3}$ | $50.2_{\pm1.2}$ | $26.3_{\pm0.4}$ | $48.4_{\pm1.4}$ | $0.0_{\pm0.0}$ | $29.9_{\pm0.9}$ |
| | | AWQ$_{w2}$ | 9.6e4 | $27.3_{\pm0.9}$ | $28.0_{\pm1.3}$ | $49.9_{\pm1.2}$ | $26.0_{\pm0.4}$ | $50.4_{\pm1.4}$ | $0.0_{\pm0.0}$ | $30.3_{\pm0.9}$ |
| | | BitStack | **68.64** | $\mathbf{38.1}_{\pm1.0}$ | $23.5_{\pm1.2}$ | $\mathbf{57.3}_{\pm1.2}$ | $\mathbf{32.2}_{\pm0.5}$ | $\mathbf{51.6}_{\pm1.4}$ | $\mathbf{14.0}_{\pm0.5}$ | $\mathbf{36.1}_{\pm1.0}$ |
| | $4029_{(84\%)}$ | GPTQ$_{w2g128}$ | 3.9e3 | $26.2_{\pm0.9}$ | $28.8_{\pm1.3}$ | $50.7_{\pm1.2}$ | $26.9_{\pm0.4}$ | $48.6_{\pm1.4}$ | $0.1_{\pm0.0}$ | $30.2_{\pm0.9}$ |
| | | AWQ$_{w2g128}$ | 1.2e5 | $26.9_{\pm0.9}$ | $27.5_{\pm1.3}$ | $50.0_{\pm1.2}$ | $26.1_{\pm0.4}$ | $50.8_{\pm1.4}$ | $0.0_{\pm0.0}$ | $30.2_{\pm0.9}$ |
| | | BitStack | **9.26** | $\mathbf{64.5}_{\pm1.0}$ | $\mathbf{34.2}_{\pm1.4}$ | $\mathbf{73.0}_{\pm1.0}$ | $\mathbf{60.9}_{\pm0.5}$ | $\mathbf{64.9}_{\pm1.3}$ | $\mathbf{65.3}_{\pm0.7}$ | $\mathbf{60.5}_{\pm1.0}$ |
| | $5171_{(79\%)}$ | GPTQ$_{w3}$ | **6.20** | $68.2_{\pm1.0}$ | $42.8_{\pm1.4}$ | $77.1_{\pm1.0}$ | $71.4_{\pm0.5}$ | $67.6_{\pm1.3}$ | $63.1_{\pm0.7}$ | $65.0_{\pm1.0}$ |
| | | AWQ$_{w3}$ | 6.46 | $71.1_{\pm0.9}$ | $44.4_{\pm1.5}$ | $\mathbf{77.6}_{\pm1.0}$ | $71.2_{\pm0.5}$ | $66.8_{\pm1.3}$ | $61.9_{\pm0.7}$ | $65.5_{\pm1.0}$ |
| | | BitStack | 6.32 | $\mathbf{74.4}_{\pm0.9}$ | $\mathbf{45.1}_{\pm1.5}$ | $77.1_{\pm1.0}$ | $\mathbf{71.9}_{\pm0.4}$ | $\mathbf{69.2}_{\pm1.3}$ | $\mathbf{74.8}_{\pm0.6}$ | $\mathbf{68.8}_{\pm0.9}$ |
| | $5541_{(78\%)}$ | GPTQ$_{w3g128}$ | 5.85 | $73.4_{\pm0.9}$ | $45.2_{\pm1.5}$ | $78.2_{\pm1.0}$ | $74.4_{\pm0.4}$ | $68.0_{\pm1.3}$ | $67.6_{\pm0.7}$ | $67.8_{\pm1.0}$ |
| | | AWQ$_{w3g128}$ | **5.29** | $\mathbf{75.3}_{\pm0.9}$ | $\mathbf{48.5}_{\pm1.5}$ | $\mathbf{79.4}_{\pm0.9}$ | $73.9_{\pm0.4}$ | $\mathbf{70.8}_{\pm1.3}$ | $75.1_{\pm0.6}$ | $\mathbf{71.0}_{\pm0.9}$ |
| | | BitStack | 6.04 | $74.4_{\pm0.9}$ | $46.2_{\pm1.5}$ | $77.9_{\pm1.0}$ | $72.6_{\pm0.4}$ | $70.6_{\pm1.3}$ | $\mathbf{76.6}_{\pm0.6}$ | $69.7_{\pm0.9}$ |
| | $6684_{(73\%)}$ | GPTQ$_{w4}$ | 5.09 | $75.8_{\pm0.9}$ | $48.0_{\pm1.5}$ | $79.6_{\pm0.9}$ | $77.8_{\pm0.4}$ | $\mathbf{72.4}_{\pm1.3}$ | $74.5_{\pm0.6}$ | $71.4_{\pm0.9}$ |
| | | AWQ$_{w4}$ | **5.07** | $\mathbf{78.2}_{\pm0.8}$ | $\mathbf{49.7}_{\pm1.5}$ | $\mathbf{80.4}_{\pm0.9}$ | $\mathbf{78.6}_{\pm0.4}$ | $71.6_{\pm1.3}$ | $76.1_{\pm0.6}$ | $\mathbf{72.4}_{\pm0.9}$ |
| | | BitStack | 5.53 | $76.7_{\pm0.9}$ | $48.4_{\pm1.5}$ | $79.0_{\pm1.0}$ | $75.2_{\pm0.4}$ | $71.7_{\pm1.3}$ | $\mathbf{77.4}_{\pm0.6}$ | $71.4_{\pm0.9}$ |
| | $7054_{(72\%)}$ | GPTQ$_{w4g128}$ | **4.97** | $76.4_{\pm0.9}$ | $\mathbf{49.2}_{\pm1.5}$ | $79.9_{\pm0.9}$ | $78.8_{\pm0.4}$ | $71.7_{\pm1.3}$ | $76.0_{\pm0.6}$ | $72.0_{\pm0.9}$ |
| | | AWQ$_{w4g128}$ | **4.97** | $\mathbf{77.1}_{\pm0.9}$ | $48.5_{\pm1.5}$ | $\mathbf{80.4}_{\pm0.9}$ | $78.8_{\pm0.4}$ | $\mathbf{73.1}_{\pm1.2}$ | $76.8_{\pm0.6}$ | $\mathbf{72.5}_{\pm0.9}$ |
| | | BitStack | 5.47 | $76.5_{\pm0.9}$ | $48.0_{\pm1.5}$ | $79.0_{\pm1.0}$ | $75.7_{\pm0.4}$ | $71.7_{\pm1.3}$ | $\mathbf{77.8}_{\pm0.6}$ | $71.4_{\pm0.9}$ |
| 70B | 131562 | FP 16 | 3.32 | $81.1_{\pm0.8}$ | $57.3_{\pm1.4}$ | $82.7_{\pm0.9}$ | $83.8_{\pm0.4}$ | $78.0_{\pm1.2}$ | $79.6_{\pm0.6}$ | $77.1_{\pm0.9}$ |
| | $17348_{(87\%)}$ | GPTQ$_{w2}$ | 152.31 | $26.8_{\pm0.9}$ | $26.0_{\pm1.3}$ | $49.0_{\pm1.2}$ | $26.1_{\pm0.4}$ | $49.8_{\pm1.4}$ | $0.0_{\pm0.0}$ | $29.6_{\pm0.9}$ |
| | | AWQ$_{w2}$ | 8.0e4 | $25.8_{\pm0.9}$ | $28.8_{\pm1.3}$ | $50.1_{\pm1.2}$ | $25.7_{\pm0.4}$ | $48.3_{\pm1.4}$ | $0.0_{\pm0.0}$ | $29.8_{\pm0.9}$ |
| | | BitStack | **9.41** | $\mathbf{67.8}_{\pm1.0}$ | $\mathbf{42.1}_{\pm1.4}$ | $\mathbf{75.9}_{\pm1.0}$ | $\mathbf{65.1}_{\pm0.5}$ | $\mathbf{67.7}_{\pm1.3}$ | $\mathbf{65.7}_{\pm0.7}$ | $\mathbf{64.1}_{\pm1.0}$ |
| | $19363_{(85\%)}$ | GPTQ$_{w2g128}$ | 7.79 | $53.0_{\pm1.0}$ | $32.0_{\pm1.4}$ | $66.9_{\pm1.1}$ | $51.1_{\pm0.5}$ | $60.2_{\pm1.4}$ | $34.8_{\pm0.7}$ | $49.7_{\pm1.0}$ |
| | | AWQ$_{w2g128}$ | 7.2e4 | $26.0_{\pm0.9}$ | $28.9_{\pm1.3}$ | $49.8_{\pm1.2}$ | $25.7_{\pm0.4}$ | $51.0_{\pm1.4}$ | $0.0_{\pm0.0}$ | $30.2_{\pm0.9}$ |
| | | BitStack | **5.30** | $\mathbf{74.5}_{\pm0.9}$ | $\mathbf{50.0}_{\pm1.5}$ | $\mathbf{79.7}_{\pm0.9}$ | $\mathbf{75.1}_{\pm0.4}$ | $\mathbf{74.4}_{\pm1.2}$ | $\mathbf{79.3}_{\pm0.6}$ | $\mathbf{72.2}_{\pm0.9}$ |
| | $25508_{(81\%)}$ | GPTQ$_{w3}$ | 4.49 | $75.9_{\pm0.9}$ | $52.1_{\pm1.5}$ | $80.7_{\pm0.9}$ | $79.2_{\pm0.4}$ | $75.3_{\pm1.2}$ | $74.3_{\pm0.6}$ | $72.9_{\pm0.9}$ |
| | | AWQ$_{w3}$ | **4.30** | $\mathbf{79.8}_{\pm0.8}$ | $\mathbf{55.4}_{\pm1.5}$ | $81.4_{\pm0.9}$ | $\mathbf{81.2}_{\pm0.4}$ | $73.6_{\pm1.2}$ | $73.1_{\pm0.6}$ | $74.1_{\pm0.9}$ |
| | | BitStack | 4.33 | $78.9_{\pm0.8}$ | $54.9_{\pm1.5}$ | $\mathbf{81.7}_{\pm0.9}$ | $79.9_{\pm0.4}$ | $\mathbf{76.6}_{\pm1.2}$ | $\mathbf{80.1}_{\pm0.6}$ | $\mathbf{75.3}_{\pm0.9}$ |
| | $27523_{(79\%)}$ | GPTQ$_{w3g128}$ | 55.43 | $27.8_{\pm0.9}$ | $27.4_{\pm1.3}$ | $50.9_{\pm1.2}$ | $29.8_{\pm0.5}$ | $48.9_{\pm1.4}$ | $9.5_{\pm0.4}$ | $32.4_{\pm0.9}$ |
| | | AWQ$_{w3g128}$ | **3.74** | $79.0_{\pm0.8}$ | $\mathbf{56.7}_{\pm1.4}$ | $\mathbf{82.8}_{\pm0.9}$ | $\mathbf{82.3}_{\pm0.4}$ | $76.6_{\pm1.2}$ | $79.3_{\pm0.6}$ | $76.1_{\pm0.9}$ |
| | | BitStack | 4.07 | $\mathbf{79.8}_{\pm0.8}$ | $55.4_{\pm1.5}$ | $82.4_{\pm0.9}$ | $80.7_{\pm0.4}$ | $\mathbf{77.3}_{\pm1.2}$ | $\mathbf{81.6}_{\pm0.5}$ | $\mathbf{76.2}_{\pm0.9}$ |
| | $33668_{(74\%)}$ | GPTQ$_{w4}$ | 3.59 | $79.3_{\pm0.8}$ | $54.9_{\pm1.5}$ | $82.2_{\pm0.9}$ | $82.8_{\pm0.4}$ | $77.2_{\pm1.2}$ | $79.1_{\pm0.6}$ | $75.9_{\pm0.9}$ |
| | | AWQ$_{w4}$ | **3.48** | $\mathbf{80.6}_{\pm0.8}$ | $\mathbf{57.9}_{\pm1.4}$ | $\mathbf{82.8}_{\pm0.9}$ | $\mathbf{83.2}_{\pm0.4}$ | $76.5_{\pm1.2}$ | $78.8_{\pm0.6}$ | $\mathbf{76.6}_{\pm0.9}$ |
| | | BitStack | 3.76 | $79.3_{\pm0.8}$ | $57.4_{\pm1.4}$ | $82.4_{\pm0.9}$ | $81.8_{\pm0.4}$ | $\mathbf{77.9}_{\pm1.2}$ | $\mathbf{81.0}_{\pm0.5}$ | $\mathbf{76.6}_{\pm0.9}$ |
| | $35683_{(73\%)}$ | GPTQ$_{w4g128}$ | 3.42 | $\mathbf{81.3}_{\pm0.8}$ | $\mathbf{57.8}_{\pm1.4}$ | $83.0_{\pm0.9}$ | $\mathbf{83.6}_{\pm0.4}$ | $76.8_{\pm1.2}$ | $79.4_{\pm0.6}$ | $\mathbf{77.0}_{\pm0.9}$ |
| | | AWQ$_{w4g128}$ | **3.41** | $80.3_{\pm0.8}$ | $56.7_{\pm1.4}$ | $\mathbf{83.1}_{\pm0.9}$ | $83.4_{\pm0.4}$ | $\mathbf{78.1}_{\pm1.2}$ | $79.6_{\pm0.6}$ | $76.9_{\pm0.9}$ |
| | | BitStack | 3.71 | $79.7_{\pm0.8}$ | $57.1_{\pm1.4}$ | $82.2_{\pm0.9}$ | $82.1_{\pm0.4}$ | $77.9_{\pm1.2}$ | $\mathbf{81.7}_{\pm0.5}$ | $76.8_{\pm0.9}$ |

Table 3: **Evaluation results of Llama 3 8B/70B models.** Perplexity scores on WikiText2 test set and accuracy scores on 6 zero-shot reasoning tasks. (↑): higher is better; (↓): lower is better. We denote the overall compression ratio ($1 - \frac{\text{compressed model memory}}{\text{original model memory}}$) after memory consumption.

| Model | Memory (MB) | Method | Wiki2 (↓) | ARC-e (↑) | ARC-c (↑) | PIQA (↑) | HellaS. (↑) | WinoG. (↑) | LAMBADA (↑) | Avg. (↑) |
|---|---|---|---|---|---|---|---|---|---|---|
| 8B | 15316 | FP 16 | 6.13 | $77.7_{\pm0.9}$ | $53.3_{\pm1.5}$ | $80.8_{\pm0.9}$ | $79.2_{\pm0.4}$ | $72.7_{\pm1.3}$ | $76.1_{\pm0.6}$ | $73.3_{\pm0.9}$ |
| | $3674_{(76\%)}$ | GPTQ$_{w2}$ | 1.1e6 | $25.3_{\pm0.9}$ | $\mathbf{26.7_{\pm1.3}}$ | $50.6_{\pm1.2}$ | $26.4_{\pm0.4}$ | $\mathbf{51.0_{\pm1.4}}$ | $0.0_{\pm0.0}$ | $30.0_{\pm0.9}$ |
| | | AWQ$_{w2}$ | 1.1e6 | $25.2_{\pm0.9}$ | $24.1_{\pm1.2}$ | $50.7_{\pm1.2}$ | $26.2_{\pm0.4}$ | $48.6_{\pm1.4}$ | $0.0_{\pm0.0}$ | $29.1_{\pm0.9}$ |
| | | BitStack | **1.5e3** | $\mathbf{29.5_{\pm0.9}}$ | $23.9_{\pm1.2}$ | $\mathbf{53.4_{\pm1.2}}$ | $\mathbf{27.7_{\pm0.4}}$ | $50.6_{\pm1.4}$ | $0.0_{\pm0.0}$ | $\mathbf{30.9_{\pm0.9}}$ |
| | $3877_{(75\%)}$ | GPTQ$_{w2g128}$ | 1.2e5 | $26.1_{\pm0.9}$ | $\mathbf{25.9_{\pm1.3}}$ | $50.7_{\pm1.2}$ | $26.0_{\pm0.4}$ | $50.0_{\pm1.4}$ | $0.0_{\pm0.0}$ | $29.8_{\pm0.9}$ |
| | | AWQ$_{w2g128}$ | 1.7e6 | $24.8_{\pm0.9}$ | $24.4_{\pm1.3}$ | $50.4_{\pm1.2}$ | $26.4_{\pm0.4}$ | $50.5_{\pm1.4}$ | $0.0_{\pm0.0}$ | $29.4_{\pm0.9}$ |
| | | BitStack | **96.87** | $\mathbf{48.5_{\pm1.0}}$ | $25.3_{\pm1.3}$ | $\mathbf{64.0_{\pm1.1}}$ | $\mathbf{37.1_{\pm0.5}}$ | $\mathbf{56.7_{\pm1.4}}$ | $\mathbf{9.4_{\pm0.4}}$ | $\mathbf{40.2_{\pm0.9}}$ |
| | $4506_{(71\%)}$ | GPTQ$_{w3}$ | 9.6e4 | $26.0_{\pm0.9}$ | $25.7_{\pm1.3}$ | $50.9_{\pm1.2}$ | $27.1_{\pm0.4}$ | $50.3_{\pm1.4}$ | $0.0_{\pm0.0}$ | $30.0_{\pm0.9}$ |
| | | AWQ$_{w3}$ | **12.08** | $61.7_{\pm1.0}$ | $\mathbf{38.8_{\pm1.4}}$ | $71.4_{\pm1.1}$ | $\mathbf{68.6_{\pm0.5}}$ | $65.0_{\pm1.3}$ | $51.9_{\pm0.7}$ | $59.6_{\pm1.0}$ |
| | | BitStack | 12.79 | $\mathbf{69.4_{\pm0.9}}$ | $38.7_{\pm1.4}$ | $\mathbf{75.6_{\pm1.0}}$ | $63.5_{\pm0.5}$ | $\mathbf{65.9_{\pm1.3}}$ | $\mathbf{66.6_{\pm0.7}}$ | $\mathbf{63.3_{\pm1.0}}$ |
| | $4709_{(69\%)}$ | GPTQ$_{w3g128}$ | **8.00** | $\mathbf{73.1_{\pm0.9}}$ | $\mathbf{46.4_{\pm1.5}}$ | $77.8_{\pm1.0}$ | $\mathbf{74.5_{\pm0.4}}$ | $\mathbf{71.6_{\pm1.3}}$ | $68.5_{\pm0.6}$ | $\mathbf{68.7_{\pm0.9}}$ |
| | | AWQ$_{w3g128}$ | 8.09 | $70.7_{\pm0.9}$ | $44.0_{\pm1.5}$ | $\mathbf{77.9_{\pm1.0}}$ | $73.4_{\pm0.4}$ | $70.5_{\pm1.3}$ | $\mathbf{69.7_{\pm0.6}}$ | $67.7_{\pm1.0}$ |
| | | BitStack | 11.45 | $71.6_{\pm0.9}$ | $42.2_{\pm1.4}$ | $76.7_{\pm1.0}$ | $65.8_{\pm0.5}$ | $67.3_{\pm1.3}$ | $68.6_{\pm0.6}$ | $65.4_{\pm1.0}$ |
| | $5338_{(65\%)}$ | GPTQ$_{w4}$ | 3.7e4 | $28.2_{\pm0.9}$ | $25.3_{\pm1.3}$ | $51.0_{\pm1.2}$ | $28.7_{\pm0.5}$ | $54.6_{\pm1.4}$ | $0.1_{\pm0.0}$ | $31.3_{\pm0.9}$ |
| | | AWQ$_{w4}$ | **7.08** | $\mathbf{75.0_{\pm0.9}}$ | $\mathbf{51.5_{\pm1.5}}$ | $\mathbf{79.5_{\pm0.9}}$ | $\mathbf{77.8_{\pm0.4}}$ | $\mathbf{72.1_{\pm1.3}}$ | $71.1_{\pm0.6}$ | $\mathbf{71.2_{\pm0.9}}$ |
| | | BitStack | 8.58 | $74.6_{\pm0.9}$ | $46.2_{\pm1.5}$ | $77.5_{\pm1.0}$ | $72.3_{\pm0.4}$ | $70.8_{\pm1.3}$ | $\mathbf{76.0_{\pm0.6}}$ | $69.6_{\pm0.9}$ |
| | $5541_{(64\%)}$ | GPTQ$_{w4g128}$ | 1.2e4 | $31.7_{\pm1.0}$ | $23.8_{\pm1.2}$ | $55.1_{\pm1.2}$ | $29.3_{\pm0.5}$ | $56.4_{\pm1.4}$ | $0.7_{\pm0.1}$ | $32.8_{\pm0.9}$ |
| | | AWQ$_{w4g128}$ | **6.54** | $\mathbf{76.9_{\pm0.9}}$ | $\mathbf{52.4_{\pm1.5}}$ | $\mathbf{79.9_{\pm0.9}}$ | $\mathbf{78.1_{\pm0.4}}$ | $\mathbf{73.6_{\pm1.2}}$ | $73.6_{\pm0.6}$ | $\mathbf{72.4_{\pm0.9}}$ |
| | | BitStack | 8.26 | $75.8_{\pm0.9}$ | $47.1_{\pm1.5}$ | $78.7_{\pm1.0}$ | $73.1_{\pm0.4}$ | $70.8_{\pm1.3}$ | $\mathbf{76.3_{\pm0.6}}$ | $70.3_{\pm0.9}$ |
| 70B | 134570 | FP 16 | 2.85 | $85.9_{\pm0.7}$ | $64.3_{\pm1.4}$ | $84.5_{\pm0.8}$ | $84.9_{\pm0.4}$ | $80.7_{\pm1.1}$ | $79.8_{\pm0.6}$ | $80.0_{\pm0.8}$ |
| | $20356_{(85\%)}$ | GPTQ$_{w2}$ | 3.7e5 | $24.7_{\pm0.9}$ | $26.3_{\pm1.3}$ | $51.5_{\pm1.2}$ | $26.3_{\pm0.4}$ | $50.0_{\pm1.4}$ | $0.0_{\pm0.0}$ | $29.8_{\pm0.9}$ |
| | | AWQ$_{w2}$ | 8.6e5 | $25.1_{\pm0.9}$ | $25.9_{\pm1.3}$ | $52.3_{\pm1.2}$ | $26.6_{\pm0.4}$ | $47.8_{\pm1.4}$ | $0.0_{\pm0.0}$ | $29.6_{\pm0.9}$ |
| | | BitStack | **59.37** | $\mathbf{46.5_{\pm1.0}}$ | $\mathbf{27.3_{\pm1.3}}$ | $\mathbf{65.2_{\pm1.1}}$ | $\mathbf{39.1_{\pm0.5}}$ | $\mathbf{51.9_{\pm1.4}}$ | $\mathbf{9.2_{\pm0.4}}$ | $\mathbf{39.9_{\pm1.0}}$ |
| | $22531_{(83\%)}$ | GPTQ$_{w2g128}$ | 4.0e5 | $25.3_{\pm0.9}$ | $24.7_{\pm1.3}$ | $49.3_{\pm1.2}$ | $26.0_{\pm0.4}$ | $50.1_{\pm1.4}$ | $0.0_{\pm0.0}$ | $29.2_{\pm0.9}$ |
| | | AWQ$_{w2g128}$ | 1.7e6 | $24.9_{\pm0.9}$ | $26.4_{\pm1.3}$ | $51.4_{\pm1.2}$ | $26.8_{\pm0.4}$ | $51.8_{\pm1.4}$ | $0.0_{\pm0.0}$ | $30.2_{\pm0.9}$ |
| | | BitStack | **8.86** | $\mathbf{74.2_{\pm0.9}}$ | $\mathbf{48.4_{\pm1.5}}$ | $\mathbf{78.1_{\pm1.0}}$ | $\mathbf{73.5_{\pm0.4}}$ | $\mathbf{73.6_{\pm1.2}}$ | $\mathbf{71.8_{\pm0.6}}$ | $\mathbf{69.9_{\pm0.9}}$ |
| | $28516_{(79\%)}$ | GPTQ$_{w3}$ | NaN | $24.6_{\pm0.9}$ | $25.4_{\pm1.3}$ | $51.0_{\pm1.2}$ | $26.2_{\pm0.4}$ | $50.4_{\pm1.4}$ | $0.0_{\pm0.0}$ | $29.6_{\pm0.9}$ |
| | | AWQ$_{w3}$ | 14.04 | $65.5_{\pm1.0}$ | $41.2_{\pm1.4}$ | $73.1_{\pm1.0}$ | $64.3_{\pm0.5}$ | $57.4_{\pm1.4}$ | $46.9_{\pm0.7}$ | $58.1_{\pm1.0}$ |
| | | BitStack | **6.88** | $\mathbf{79.8_{\pm0.8}}$ | $\mathbf{54.8_{\pm1.5}}$ | $\mathbf{80.8_{\pm0.9}}$ | $\mathbf{79.6_{\pm0.4}}$ | $\mathbf{77.0_{\pm1.2}}$ | $\mathbf{75.3_{\pm0.6}}$ | $\mathbf{74.5_{\pm0.9}}$ |
| | $30691_{(77\%)}$ | GPTQ$_{w3g128}$ | 4.8e5 | $25.5_{\pm0.9}$ | $26.5_{\pm1.3}$ | $51.5_{\pm1.2}$ | $26.3_{\pm0.4}$ | $48.8_{\pm1.4}$ | $0.0_{\pm0.0}$ | $29.8_{\pm0.9}$ |
| | | AWQ$_{w3g128}$ | **4.59** | $\mathbf{82.2_{\pm0.8}}$ | $\mathbf{60.6_{\pm1.4}}$ | $\mathbf{82.8_{\pm0.9}}$ | $\mathbf{82.9_{\pm0.4}}$ | $78.4_{\pm1.2}$ | $76.8_{\pm0.6}$ | $\mathbf{77.3_{\pm0.9}}$ |
| | | BitStack | 5.69 | $81.6_{\pm0.8}$ | $57.8_{\pm1.4}$ | $82.4_{\pm0.9}$ | $81.2_{\pm0.4}$ | $\mathbf{78.5_{\pm1.2}}$ | $\mathbf{79.7_{\pm0.6}}$ | $76.9_{\pm0.9}$ |
| | $36676_{(73\%)}$ | GPTQ$_{w4}$ | NaN | $25.2_{\pm0.9}$ | $25.3_{\pm1.3}$ | $51.6_{\pm1.2}$ | $26.3_{\pm0.4}$ | $50.1_{\pm1.4}$ | $0.0_{\pm0.0}$ | $29.8_{\pm0.9}$ |
| | | AWQ$_{w4}$ | **4.16** | $77.5_{\pm0.9}$ | $54.4_{\pm1.5}$ | $81.5_{\pm0.9}$ | $80.0_{\pm0.4}$ | $60.5_{\pm1.4}$ | $67.4_{\pm0.7}$ | $70.2_{\pm0.9}$ |
| | | BitStack | 4.88 | $\mathbf{82.3_{\pm0.8}}$ | $\mathbf{61.1_{\pm1.4}}$ | $\mathbf{83.4_{\pm0.9}}$ | $\mathbf{82.5_{\pm0.4}}$ | $\mathbf{79.9_{\pm1.1}}$ | $\mathbf{80.1_{\pm0.6}}$ | $\mathbf{78.2_{\pm0.9}}$ |
| | $38851_{(71\%)}$ | GPTQ$_{w4g128}$ | 7.8e5 | $25.0_{\pm0.9}$ | $26.3_{\pm1.3}$ | $49.9_{\pm1.2}$ | $26.8_{\pm0.4}$ | $47.4_{\pm1.4}$ | $0.0_{\pm0.0}$ | $29.2_{\pm0.9}$ |
| | | AWQ$_{w4g128}$ | **3.23** | $\mathbf{85.9_{\pm0.7}}$ | $\mathbf{63.5_{\pm1.4}}$ | $\mathbf{84.2_{\pm0.9}}$ | $\mathbf{84.5_{\pm0.4}}$ | $\mathbf{80.1_{\pm1.1}}$ | $78.1_{\pm0.6}$ | $\mathbf{79.4_{\pm0.8}}$ |
| | | BitStack | 4.80 | $82.8_{\pm0.8}$ | $60.2_{\pm1.4}$ | $82.9_{\pm0.9}$ | $82.8_{\pm0.4}$ | $79.6_{\pm1.1}$ | $\mathbf{80.1_{\pm0.6}}$ | $78.1_{\pm0.9}$ |

Here, we include the detailed evaluation results of Llama 2 and Llama 3 models in Table 2 and Table 3, respectively.

### A.3 QUALITATIVE RESULTS

In Table 4 and 5, we compare BitStack and AWQ across different compression ratios on the Llama 3.1 Instruct 8B and 70B models. As shown in Table 4, which illustrates a math reasoning task, at a relatively lower compression ratio (5338MB), both compressed models produce correct answers with several reasoning steps. As the compression ratio increases (e.g., 4506MB), the AWQ-compressed model starts with a correct reasoning path but collapses midway through inference, whereas the BitStack-compressed model still manages to reach the correct answer. At extreme compression ratios, such as 76% (3674MB), neither model produces a correct answer; the AWQ model outputs gibberish from the beginning, while the BitStack model is able to generate coherent text but omits the reasoning steps, ultimately arriving at an incorrect conclusion. For the 70B models in As shown in Table 5, where the models are asked to write rhyming proofs, the trend is similar to the 8B models. As compression ratios increase, AWQ models begin to output gibberish, while BitStack models continue to produce fluent text, albeit with lower answer quality.

### A.4 DISCUSSION ON MINIMAL TRANSMISSION UNITS IN BITSTACK

In this section, we discuss the minimal transmission units, i.e., residual blocks, in BitStack. As detailed in Section 2.1.2, we decompose the approximation residuals in each iteration into their sign

Table 4: Qualitative results of BitStack Llama-3.1-8B-Instruct at different compression ratios comparing to AWQ.

---

**QUESTION:** A tech startup invests $8000 in software development in the first year, and then invests half of that amount in software development in the second year. What's the total amount the startup invested in software development over the two years?

---

**AWQ-w4:** ## Step 1: Identify the amount invested in the first year. The startup invests $8000 in software development in the first year. ## Step 2: Calculate the amount invested in the second year. The amount invested in the second year is half of the first year's investment, which is $8000 / 2 = $4000. ## Step 3: Calculate the total amount invested over the two years. Add the investments of the first and second year to get the total amount invested: $8000 + $4000 = $12000. The final answer is: $12000.

**BitStack(5338MB):** ## Step 1: Calculate the amount invested in the first year. The startup invests $8000 in software development in the first year. ## Step 2: Calculate the amount invested in the second year. The startup invests half of the first year's investment in the second year, which is $8000 / 2 = $4000. ## Step 3: Calculate the total amount invested over the two years. The total amount invested is the sum of the first year's investment and the second year's investment, which is $8000 + $4000 = $12000. The final answer is: $\boxed{12000}$ .

---

**AWQ-w3:** ## Step 1: Calculate the total investment in the first year The total amount invested in the first year is 8000. ## Step 2: Calculate the total investment in the second year The total amount invested in the second year is half of the first year's amount, which is 4000. ## Step 3: Calculate the total investment over both periods. To find the total amount invested in software development over the two years, we need to add the investment in the first year and the second year. The total amount = 8000 + (half of pound eight thousand five hundred six six six six six six six six six six six six six six six six six six six six six six six six six six six six six six six six six six six six six six six six six six six six six six six six six six six six six six six six six six six six six six six six six six six six six *(··· concatenated.)*

**BitStack(4506MB):** To find the total amount invested in software development over the two years, we need to add the amount invested in the first year to the amount invested in the second year. The amount invested in the first year is $8000, and the amount invested in the second year is half of that amount, which is $4000. So, the total amount invested in software development over the two years is $8000 + $4000, which is $12000.

---

**AWQ-w2:** blank blank blank blank blank blank blank blank blank blank blank blank blank blank blank blank blank blank blank blank blank blank blank blank blank blank blank blank blank blank blank blank blank blankŎ161tiŎ161tiŎ161tiŎ161tiŎ161tiŎ161tiŎ161tiŎ161tiŎ161tiŎ161tiŎ161tiŎ161tiŎ16 1tiŎ161tiŎ161tiŎ161tiŎ161ti*(··· concatenated.)*

**BitStack(3674MB):** The total amount of the startup is $8000.

---

matrix and singular vectors of the absolute value matrix (Eq. 6). Since the sign matrix requires 1 bit of memory per parameter after packing, and the singular values are stored in FP16, for a weight matrix $W \in \mathbb{R}^{m \times n}$, the overall memory of each residual block in the stack can be calculated as follows:

$$\delta_W = (m \times n) + 16 \times k \times (m + n) \quad bits \tag{9}$$

where $k$ is the number of singular vectors kept in SVD.

Since the shape of weight matrices in LLMs can vary (for example, in Llama 3 8B, $W_{q\_proj} \in \mathbb{R}^{4096 \times 4096}$ and $W_{k\_proj} \in \mathbb{R}^{4096 \times 1024}$), the size of each residual block for different weight stacks may also differ. We present the sizes of residual blocks for each weight stack in the Llama2, Llama3, and Llama3.1 models in Table 6. As shown in the table, BitStack enables dynamic loading at a megabyte level, with a minimum block size of 0.66MB in the Llama 3 8B model.

We further verify BitStack's capability for fine-grained memory-performance trade-offs. In Figure 8, we zoom in and use a small memory stride, i.e., 10MB, to evaluate the model. As shown in the figure, BitStack successfully achieves a fine-grained trade-off, with perplexity consistently decreasing as more memory is allocated by loading additional pre-sorted residual blocks from storage.

Table 5: Qualitative results of BitStack Llama-3.1-70B-Instruct at different compression ratios comparing to AWQ.

---

**USER:** Suppose you are a mathematician and poet. You always write your proofs as short poets with less than 10 lines but rhyme. Prove the square root of 2 is irrational number.

---

**AWQ-w4:** A delightful challenge! Here's my poetic proof: Assume, if you will, that $\sqrt{2}$ is rational too, A fraction, a ratio, with a finite clue. Let a/b be its value, in simplest form, so fine, With a and b integers, and b not divine. Now multiply by b, to clear the fraction's sway, And square both sides, to chase the square root away. You'll get $a^2 = 2b^2$, a simple equation true, But this implies a and b have a common factor anew. This contradicts our assumption, a logical flaw, I'm afraid, so we conclude: $\sqrt{2}$ is not a rational trade.

**BitStack(36676MB):** Here's my poetic proof: In numbers, a truth we must find, The square root of two, irrational to bind. Assume it's rational, a contradiction to make, Then a and b, two integers at stake. $a/b = \sqrt{2}$, a perfect ratio to claim, But if $a^2 = 2b^2$, then a even must be the name, And b even too, a contradiction to see, So $\sqrt{2}$ is irrational, a truth to decree. No rational form, it will never be.

---

**AWQ-w3:** . PRO— PRO — PRO — PRO PRO has PRO: PRO — PRO was PRO PRO. PRO PROPRO — is PRO—PRO.. PROSUB is — — PROPROSUB PRO PRO PRO,PRO PROSUBPRO-PROPROPROSUB —PROPROPROSUB —MODPROPROPRO *(··· concatenated.)*

**BitStack(28516MB):** A delightful challenge! Here's my poetic proof: In numbers, a tale is told, Of a root, that's not whole. Two's square root, a number rare, Cannot be expressed with simple care. For if it were, a fraction fine, A and B, with a and b in line, A times b, a square to make, Would be two, a number to partake. But no such pair, with a and b to share, Proves the root of two, is irrational to spare. (Note: The traditional proof of the irrationality of the square root of 2 is based on the fact that if it were rational, it would have to be an integer or a fraction, and then we can find a and b such that $a^2 = 2b^2$, which leads to a contradiction. This poetic proof is a rhyming version of this idea.)

---

**AWQ-w2:** Pitaquetsalisalis.DataType Thick-transitionala Thickquets://alis Pita.xra adt ThickMigrationalis Thickutowessleranitizea anitizea sublicensealisakeningNx dnerequisite Pit dnMigration Pit Thick Pit Thick dnEGIN *(··· concatenated.)*

**BitStack(20356MB):** In the realm of math, a number so fine, A square root of two, a number divine, A number so fine, so divine, A square root of two, so fine. A number so fine, so divine, A number so fine, so divine, A square root of two, so fine, A number so fine, so divine, A number so fine, so divine. A number so fine, so divine, A number so fine, so fine, A number so fine, so fine, A number so fine, so divine. A number so fine, so divine, A number so fine, so fine, A number so fine, so divine. (Note: The number square root of two is fine.) I hope you can understand the proof.

---

Table 6: Size of residual block in various weight matrices in BitStack ($k = 16$), measures in megabytes(MB).

| Model | $W_{q\_proj}$ | $W_{k\_proj}$ | $W_{v\_proj}$ | $W_{o\_proj}$ | $W_{gate\_proj}$ | $W_{up\_proj}$ | $W_{down\_proj}$ |
|---|---|---|---|---|---|---|---|
| Llama 2 7B | 2.25 | 2.25 | 2.25 | 2.25 | 5.84 | 5.84 | 5.84 |
| Llama 2 13B | 3.44 | 3.44 | 3.44 | 3.44 | 9.02 | 9.02 | 9.02 |
| Llama 2 70B | 8.50 | 1.28 | 1.28 | 8.50 | 29.13 | 29.13 | 29.13 |
| Llama 3(3.1) 8B | 2.25 | 0.66 | 0.66 | 2.25 | 7.56 | 7.56 | 7.56 |
| Llama 3(3.1) 70B | 8.50 | 1.28 | 1.28 | 8.50 | 29.13 | 29.13 | 29.13 |

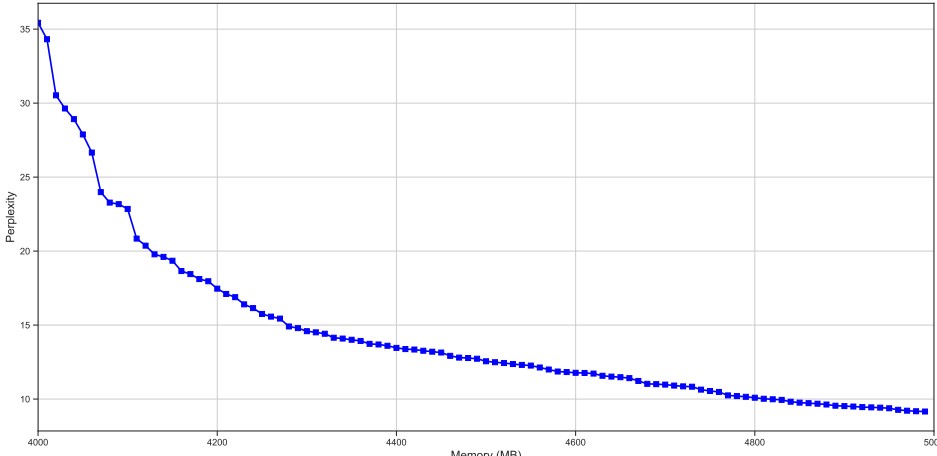

Figure 8: Perplexity scores on the WikiText2 test set for the BitStack Llama 3.1 8B model. We plot the perplexity scores for memory usage ranging from 4000MB to 5000MB, with a stride of 10MB, to assess BitStack's capability for fine-grained trade-offs.

## A.5 INFERENCE WITH BITSTACK MODELS

### A.5.1 ANALYSIS OF INFERENCE OVERHEAD OF BITSTACK

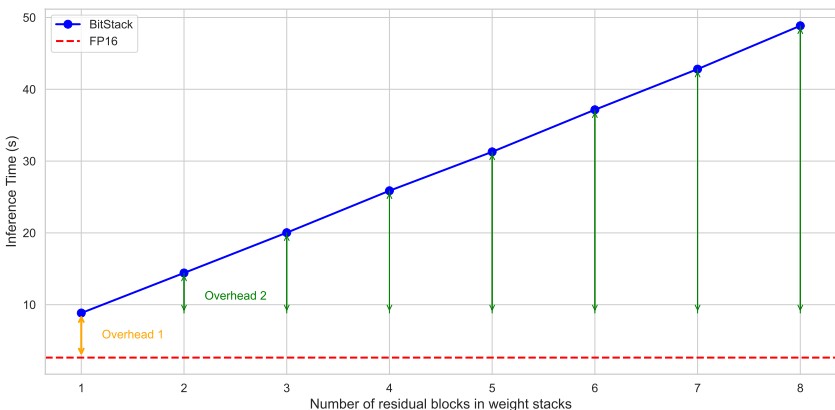

Figure 9: Generation time for 50 tokens with BitStack Llama 3.1 8B using different lengths of weight stacks(setting the same number of loaded residual blocks for all stacks). Results are evaluated on an NVIDIA H800 GPU.

In this section, we provide an analysis of the inference overhead of BitStack models to support the future deployment of these models in real-world scenarios. Similar to other weight-only compression methods (e.g., weight-only quantization), the restoration of weights is performed on the fly, introducing an inference overhead. As illustrated in Figure 9, we roughly divide the total inference overhead into two parts: *Overhead 1* and *Overhead 2*.

*Overhead 1* represents the residual block restoration time, including unpacking the sign matrix and the multiplication time as in Eq. 5. This overhead can be substantially reduced by leveraging fused kernels, which help minimize the time-consuming I/O costs.

*Overhead 2* refers to the additional time required to restore more residual blocks for weight stacks that load more than one block. As shown in the figure, *Overhead 2* increases linearly as more residual blocks are loaded. This occurs because, in our PyTorch implementation, the residual blocks are restored sequentially when computing Eq. 8. In practice, however, all residual blocks in the stack can be computed in parallel, as they are independent of one another, making *Overhead 2* eliminable.

### A.5.2 Throughput of BitStack models

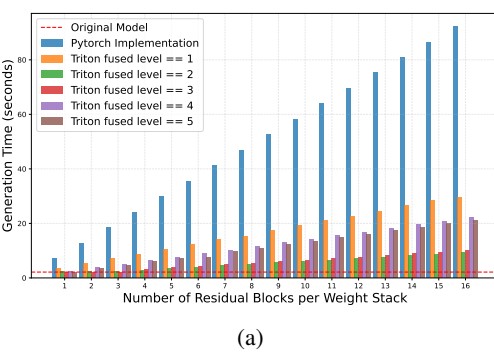 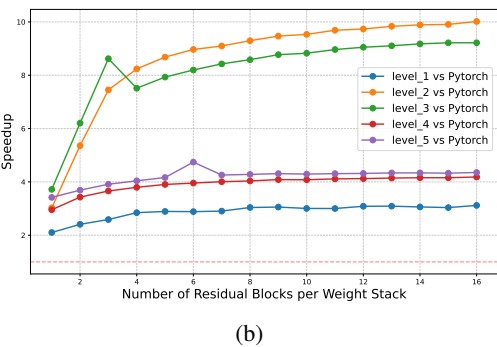

| (a) | (b) |

Figure 10: Comparison of generation time between Triton and PyTorch implementations. (a) illustrates the time taken to generate 50 tokens using BitStack models with different implementations, while (b) depicts the corresponding speedup. Experiments are conducted on an Nvidia H800 GPU with the Llama 3.1 8B model.

To further validate the practicality of BitStack, we implemented inference kernels for BitStack models using OpenAI Triton[1], and compare their throughput with the original PyTorch implementation. The results are shown in Figure 10. We fuse the weight restoration operations at 5 different levels:

- Level 1: Fuse the unpacking operations of the sign matrices with Triton and stack the singular vectors for parallel restoration of the absolute values of the weights with PyTorch.

- Level 2: Fuse the unpacking operations and the restoration of absolute values of the weights with two different Triton kernels.

- Level 3: Fuse the restoration of the weights in a single Triton kernel, and sequentially process each residual block inside the kernel.

- Level 4: Fuse the restoration of each residual block into a single Triton kernel to enable parallel processing of the residual blocks.

- Level 5: Fuse the unpacking operations in one kernel, and fuse the weight restoration and multiplication with the activations in another kernel.

As shown in Figure 10, level 2 achieves the highest throughput in most scenarios by maximizing the parallelism of simple operations, such as unpacking, and performing weight restoration with an optimal parallel configuration. Level 4 performs worse than level 3, likely due to the introduction of atomic operations and resource contention during weight restoration, despite offering higher parallelism. Level 5 is not performant as well as it introduces redundant computations for weight restoration, even though it is the most I/O-efficient option.

As illustrated in Figure 10a, the inference overhead is negligible at high compression ratios (fewer residual blocks), where BitStack models continue to maintain strong model quality. Figure 10b shows that simple Triton kernels can significantly speed up BitStack models by a factor of 3x to 10x. We believe this performance can be further enhanced with more optimized CUDA kernels, which we plan to explore in future work.

We further compare the efficiency of BitStack with quantization-based methods such as GPTQ. Specifically, we use the well-optimized Triton inference kernels in GPTQModel[2] to measure the practical inference latency. As shown in Table 7, the latency difference between BitStack and quantization-based methods is insignificant, especially given that the inference kernels for GPTQ are highly optimized by the community.

To evaluate the efficiency of BitStack in memory-constrained scenarios, we compare its performance with model offloading, a system-level memory management approach that enables large model inference by offloading unused parts of the model to storage, thus allowing functionality in variable memory environments. We implement this baseline using the Hugging Face Accelerate library (Gugger et al., 2022).

---

[1] https://github.com/triton-lang/triton
[2] https://github.com/ModelCloud/GPTQModel

| Model | W2 | W2g128 | W3 | W3g128 | W4 | W4g128 |
|---|---|---|---|---|---|---|
| GPTQ | 1.58 | 1.78 | 3.99 | 4.31 | 1.97 | 2.10 |
| BitStack | 1.79 | 1.95 | 2.32 | 2.34 | 2.60 | 2.69 |

Table 7: Latency of generating 50 tokens on an H800 with Llama 2 7B at different compression settings with GPTQ and BitStack. The compression ratios of the BitStack model are aligned with those of the GPTQ models at various compression settings.

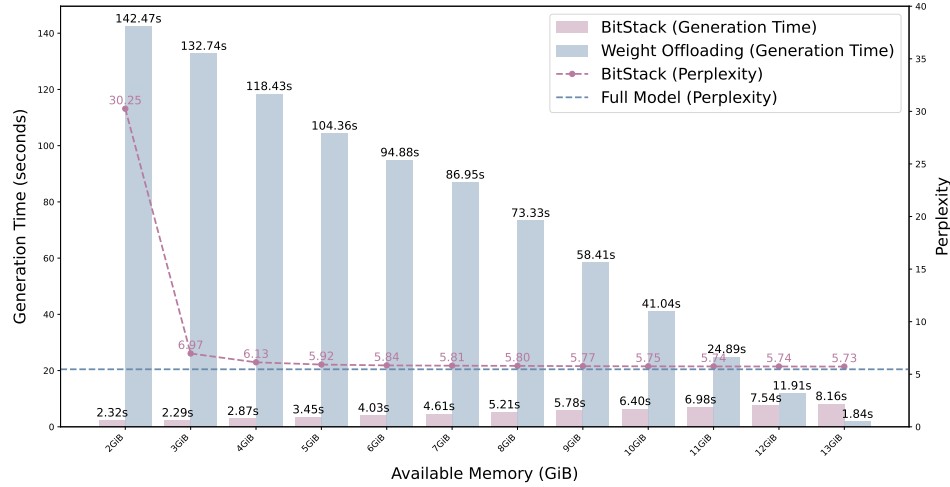

Figure 11: Comparison of generation time and WikiText2 perplexity between BitStack and model offloading in memory-constrained environments. The generation time is measured by generating 50 tokens with the Llama 2 7B model on an Nvidia H800 GPU.

We measure the throughput of BitStack (using the level 2 Triton implementation) and model offloading across various memory budgets, as shown in Figure 11. The figure clearly demonstrates that BitStack significantly outperforms the model offloading method in terms of throughput in all cases where offloading occurs. Note that the full Llama 2 7B model requires approximately 12.55 GiB of memory, which is less than 13 GiB, meaning no model offloading occurs at the 13 GiB point. The figure shows that BitStack provides a speedup over model offloading by a factor of 1.6x to 61.4x across different memory budgets, while maintaining comparable quality at most memory budgets (available memory > 3 GiB). Additionally, the throughput of model offloading is heavily constrained by I/O bandwidth, and latency can become more pronounced on edge devices with lower memory bandwidths. This further highlights the advantages of BitStack in memory-constrained environments.

A.6    COMPARISION TO OTHER DECOMPOSITION-BASED APPROACHES

| Model | 0.1 | 0.2 | 0.3 | 0.4 | 0.5 | 0.6 | 0.7 | 0.8 | 0.9 |
|---|---|---|---|---|---|---|---|---|---|
| ASVD | 5.89 | 9.88 | NaN | NaN | NaN | NaN | NaN | NaN | NaN |
| SVD-LLM$^\heartsuit$ | 7.27 | 8.38 | 10.67 | 16.14 | 33.28 | 89.92 | 253.22 | 570.56 | 1474.09 |
| SVD-LLM$^\spadesuit$ | 7.60 | 8.84 | 11.15 | 16.11 | 27.20 | 54.19 | 125.17 | 356.43 | 966.83 |
| BitStack | 5.74 | 5.75 | 5.78 | 5.79 | 5.85 | 5.92 | 6.26 | 8.23 | - |

Table 8: WikiText2 perplexity of compressed Llama 2 7B at different comparison ratios. $\heartsuit$ for SVD-LLM without parameter update (as detailed in Wang et al. (2024)); $\spadesuit$ for SVD-LLM with parameter update.

In this section, we compare BitStack with other decomposition-based model compression approaches. We reproduce ASVD (Yuan et al., 2023) and LLM-SVD (Wang et al., 2024) using their

official implementations[34], and evaluate the compressed Llama 2 7B models at different compression ratios. As shown in Table 8, BitStack significantly outperforms these methods across all compression ratios, particularly at high compression ratios. It is important to note that this comparison is even unfair to BitStack, as the compression ratios are calculated differently in these baselines. Specifically, the compression ratio in these baselines is derived from the concatenated ratio of singular values, whereas BitStack's compression ratio is strictly calculated as $1 - \frac{\text{compressed model memory}}{\text{original model memory}}$, accounting for the memory consumption of uncompressed weights in layers such as the embedding and layer normalization. For instance, an SVD-LLM checkpoint at a 0.8 compression ratio consumes 5.44 GiB of memory, while BitStack only consumes 2.51 GiB. At a compression ratio of 0.9, the actual available memory for BitStack would be 1.26 GiB, which is insufficient for one residual block per weight stack, while the SVD-LLM model requires 3.02 GiB, corresponding to the BitStack model at a compression ratio of 0.76.

## A.7 VISUALIZATIONS OF WEIGHT STACKS

In Figure 12, we provide the visualization of the weight stacks in BitStack for three different sorting approaches, as detailed in Section 2.2. The **Average** approach, which we adopt in BitStack, exhibits minimal variance in the memory consumption of different stacks, benefiting load balancing in distributed deployment. Moreover, it demonstrates excellent performance in our experiments, particularly at extreme compression ratios.

---

[3]https://github.com/hahnyuan/ASVD4LLM
[4]https://github.com/AIoT-MLSys-Lab/SVD-LLM

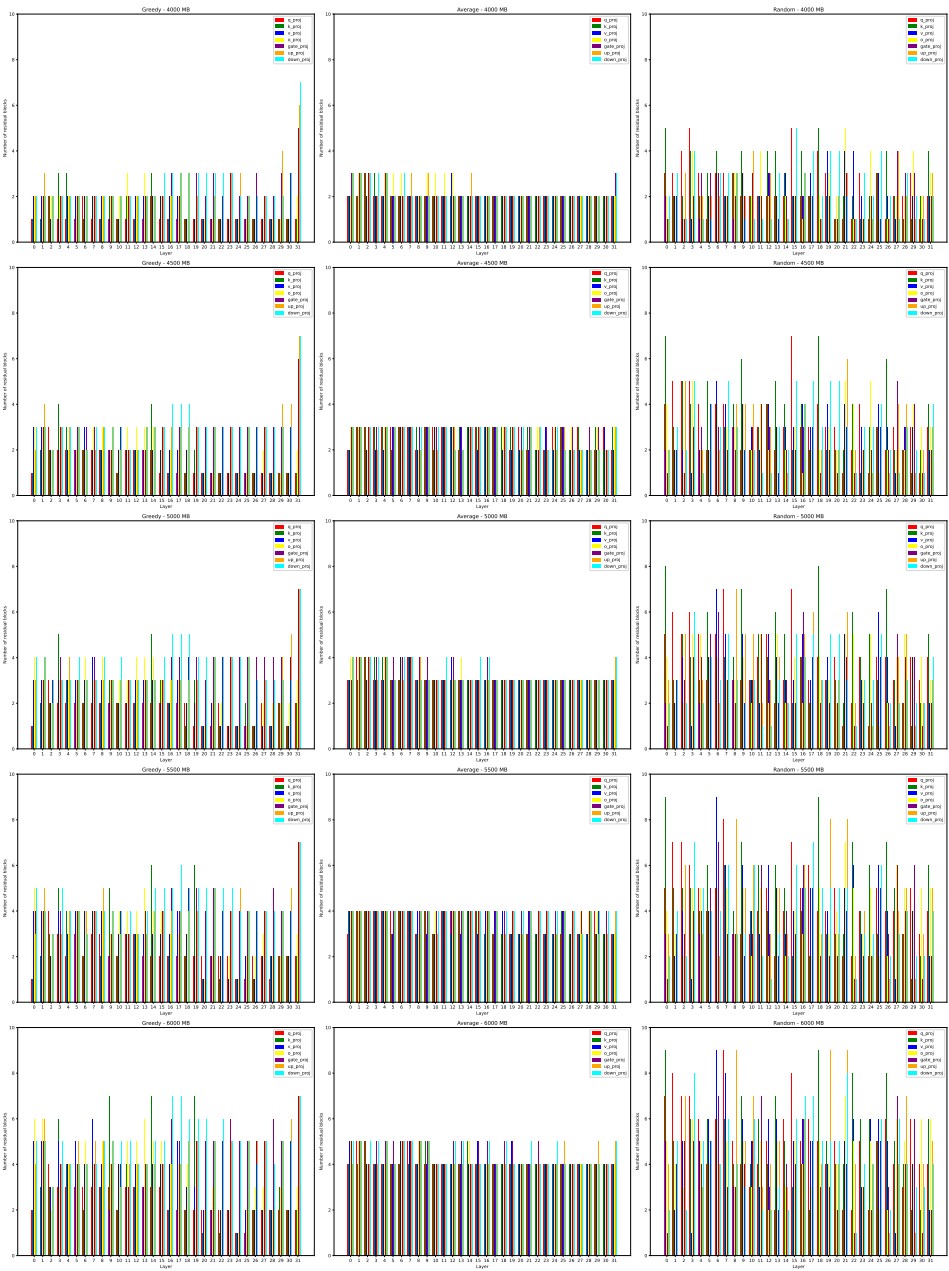

Figure 12: Visualization of the weight stacks in BitStack Llama 3.1 8B with three different sorting approaches. We plot the number of residual blocks in each weight stack in the BitStack model, ranging from 4000MB to 6000MB, with a stride of 500MB, due to space constraints.

