# OpenReview forum: "BitStack: Any-Size Compression of Large Language Models in Variable Memory Environments"
_ICLR.cc/2025/Conference — ICLR 2025 Poster_

### Official Review · Reviewer_GmCn · 2024-10-20

**Soundness:** 2
**Presentation:** 2
**Contribution:** 2
**Rating:** 6
**Confidence:** 4

**Summary:**

This paper presents BitStack, a training-free weight compression method for large language models that adeptly navigates memory constraints on local devices. By utilizing weight decomposition and stacking blocks in storage as basic transmission units, BitStack enables dynamic adjustments to model size, offering megabyte-level trade-offs between memory usage and performance. The method achieves comparable or superior results to traditional quantization techniques.

**Strengths:**

- The paper addresses a crucial problem in memory allocation for large language models (LLMs) in edge computing scenarios with limited RAM.
- The research is backed by solid experimental evidence, enhancing its credibility.

**Weaknesses:**

- The writing could be improved for clarity. The rationale for sorting SVD matrices is not well-explained, and the process of loading/offloading within and between layers needs better elucidation.
- The paper doesn't provide details about the caching process, which seems to be an integral part of the method. Discussion on cache size and caching strategy is missing.
- The impacts of SVD on model performance aren't discussed. The paper should elaborate on the effects of various decomposition parameters and distinguish the proposed SVD from those used in previous model compression techniques.
- The sorting algorithm and strategy need better description and justification. It's unclear how different matrices in LLMs should be decomposed and sorted using the proposed method.

The rebuttal has solved most of my concerns therefore I'd like to raise my score. However, I still think the scenario defined in this paper can be changed a bit. As the author claimed in their response, "BitStack is proposed as a general weight compression method for LLMs".  If the paper emphasize it is well suited for mobile/edge scenarios, I'd like to see more system design involved.

**Questions:**

- Certain terms, like 'residual block', need to be defined more clearly as it seems to differ from its usage in the context of CNNs.
- The choice of "zeropoint quantization" as a baseline needs to be justified since simple zero point quantization is known to be ineffective in LLMs.

---

> ### Author Response · Authors · 2024-11-18
> **Response to Reviewer GmCn (1/2)**
>
> As the authors, we sincerely appreciate any time and effort the reviewer has taken to improve our work and are committed to addressing any concerns about the paper to the best of our ability. However, **we strongly suspect that this review may have been generated by an LLM, as many of the questions raised appear to be based on misunderstandings or are even irrelevant to our proposed method.** Nevertheless, we will do our best to address them here:
>
> > **Q1.A**: The rationale for sorting SVD matrices is not well-explained.
> >
>
> **A1.A**: Thank you for pointing this out, and we are willing to improve the readability of the paper. We explained the sorting process in Section 2.2 and provided the pseudocode of the sorting process in Algorithm 1, Section A.1 (also referenced in the main paper). **We would appreciate if you could specify which aspects of the sorting process require further clarification, allowing us to provide more detailed explanations.** In addition, to clarify, we sort "residual blocks," which consist of a packed sign matrix and singular vectors, as illustrated in Figure 3.
>
> > **Q1.B**:  The process of loading/offloading within and between layers needs better elucidation.
> >
>
> **A1.B**: **The process of loading/offloading residual blocks does not happen within or between layers, rather, it happens between RAM and storage devices.** As illustrated in Figure 2, the residual blocks for each weight stack are loaded/offloaded to the universal stack in the storage in a presorted order, we use the same color for blocks belonging to the same weight stack for better elucidation.
>
> > **Q2**: The paper doesn't provide details about the caching process, which seems to be an integral part of the method. Discussion on cache size and caching strategy is missing.
> >
>
> **A2**: **There is no “caching process” mentioned in any part of the paper. This question seems like a hallucination by an LLM.**
>
> > **Q3**: The impacts of SVD on model performance aren't discussed. The paper should elaborate on the effects of various decomposition parameters and distinguish the proposed SVD from those used in previous model compression techniques.
> >
>
> **A3**: Thank you for raising this point. We address your concerns as follows:
>
> - In BitStack, **model performance is not directly influenced by SVD itself** as we iteratively perform SVD on the absolute value of the approximation error of each weight matrix(as discussed in Section 2.1). **It is the number of loaded residual blocks that directly impacts model performance**, which we can control to implement a trade-off between memory consumption and performance. Figure 1a provides an overview of this trade-off, demonstrating that BitStack can recover the full potential of the original model with the same memory consumption while maintaining good performance at lower memory budgets. Additionally, Figure 8 shows that this trade-off is fine-grained, where even a small increase in memory allocation leads to performance improvements.
> - Considering the number of decomposition iterations $n$ is forward-compatible(as we perform the decomposition on residuals), **the only parameter related to the decomposition is the number of retained singular vectors $k$, which we already did the ablation in Section 3.3 (Figure 7b). The results show that BitStack is largely insensitive to this parameter.**
> - The main differences between the use of SVD in BitStack and in previous methods can be summarized as follows:
>     - 1. **BitStack does not incorporate additional processes for the weight scaling before SVD** (calculation of the Hessian matrix in FWSVD[1], whitening in SVD-LLM[2], etc.).
>     - 2. **BitStack performs SVD on the previous approximation errors**, instead of directly on the model weights.
>     - 3. **BitStack retains the sign matrix**(which can be packed for further memory reduction) and **performs SVD on the absolute value** of the approximation errors.

---

> > ### Author Response · Authors · 2024-11-18
> > **Response to Reviewer GmCn (2/2)**
> >
> > > **Q4**: The sorting algorithm and strategy need better description and justification. It's unclear how different matrices in LLMs should be decomposed and sorted using the proposed method.
> > >
> >
> > **A4**: This seems to be a duplication of Q1.
> >
> > > **Q5**: Certain terms, like 'residual block', need to be defined more clearly as it seems to differ from its usage in the context of CNNs.
> > >
> >
> > **A5**: Thank you for pointing this out. In the context of BitStack, **a "residual block" refers to a packed sign matrix derived from the previous approximation error, along with the singular vectors retained after performing SVD on the absolute values of the approximation error**. Figure 3 provides a detailed illustration of a "residual block" in BitStack. See also Equation (7) for a mathematical expression of a 'residual block'.
> >
> > > **Q6**: The choice of "zeropoint quantization" as a baseline needs to be justified since simple zero point quantization is known to be ineffective in LLMs.
> > >
> >
> > **A6**: **“zeropoint quantization” itself is not used as a baseline.** It is a quantization approach alternative to “absmax quantization”, which incorporates an extra “zeropoint” parameter in each quantization operation. **We use “zeropoint quantization” for the quantization operations in GPTQ and AWQ for stronger performance for these baselines.**
> >
> > [1] Language model compression with weighted low-rank factorization https://arxiv.org/abs/2207.00112
> >
> > [2] SVD-LLM: Truncation-aware Singular Value Decomposition for Large Language Model Compression ****https://arxiv.org/abs/2403.07378

---

> > ### Comment · Reviewer_GmCn · 2024-11-22
> > **Unwarranted critism in the rebuttal**
> >
> > I strongly suspect that the author lacks sufficient understanding of system-level research. I mentioned the caching scheme because the paper introduces an "ITERATIVE ABSOLUTE VALUE DECOMPOSITION" method to compress model weights. To recover the weights, it seems necessary to cache the intermediate results from each iteration. This is because the decision to cache LLM's weights in DRAM on mobile devices should be carefully considered, given the significantly slower data loading speeds on such platforms.
> >
> > Since the paper does not appear to account for the practical constraints of mobile environments, I recommend reconsidering the proposed application scenario and targeting more powerful devices instead. This adjustment would align better with the method's requirements and practical feasibility.

---

> ### Author Response · Authors · 2024-11-23
>
> Thank you for your response. To clarify, BitStack is proposed as a general weight compression method for LLMs, and there is no caching involved at the method level. Regarding the restoration of weights from the ITERATIVE ABSOLUTE VALUE DECOMPOSITION, we provide the  implementation here:
>
> ```python
> w = torch.zeros((out_features, in_features))
> for i in n_blocks:
> 		w += unpack(packed_sign) * (u_k @ vt_k)
> ```
> We believe there is no additional loading should be worried about since we are only performing inference computations with the blocks that are already loaded. During the recovery of the weights, the intermediate results(an accumulator) occupy the same space as the resulting weights, it is unlikely that offloading to storage would happen here(if that is your concern).
>
> Additionally, could you please confirm whether your other concerns from the previous review have been addressed? Please let us know if you have any other questions.

---

### Official Review · Reviewer_g6Mj · 2024-10-30

**Soundness:** 3
**Presentation:** 2
**Contribution:** 2
**Rating:** 6
**Confidence:** 4

**Summary:**

The paper introduces BitStack, an iterative weight decomposition method to compress the Large Language Models (LLMs). By adjusting the number of iterations, BitStack can compress LLMs with various compression ratio. Experimental results demonstrate that BitStack achieves comparable performance to quantization-based methods.

**Strengths:**

1.It is the first decomposition-based method that achieves comparable performance to quantization approaches.

2.The motivation of variable RAM on devices sounds is interesting.

3.The model compressed by BitStack is more scalable for deployment.

**Weaknesses:**

1.The evaluations should be enhanced. Both AWQ and GPTQ are no longer considered state-of-the-art quantization methods as of the current time. Even AWQ was submitted to arXiv over a year ago (June 2023).

2.Despite this, the performance gap between BitStack and AWQ is relatively small. For instance, the average accuracy scores of BitStack surpass those of AWQ-w2 by 1.4 points (30.8 vs. 29.4) across the LLaMA 3.1 8B model. Given that the full-precision model achieves an accuracy score of 74.1, this improvement is quite marginal.

3. When the compression ratio is below 70%, BitStack's performance tends to be weaker than AWQ on most benchmarks.

4. What is the real-world inference speed of BitStack? This is a crucial aspect for a compression method targeting large language models (LLMs), yet it is not addressed in the paper.

5. The font size in Figures 1, 2, 5, 6, and 7 is too small

**Questions:**

See weaknesses above.

---

> ### Author Response · Authors · 2024-11-18
> **Response to Reviewer g6Mj (1/3)**
>
> We sincerely appreciate the time and effort you took to review our paper. We are grateful for your recognition of our work, and we acknowledge that most of your concerns are focused on the evaluation aspects. Below, we address these points in detail:
>
> > **Q1**: The evaluations should be enhanced. Both AWQ and GPTQ are no longer considered state-of-the-art quantization methods as of the current time. Even AWQ was submitted to arXiv over a year ago (June 2023).
> >
>
> **A1**: We acknowledge that both AWQ and GPTQ were proposed over a year ago, and they may no longer be considered state-of-the-art quantization-based approaches. However, we would like to emphasize several important reasons for selecting these methods as baselines:
>
> 1. **Training-free setting**: We consider the experiments in a training-free setting, which is a **key advantage for real-world deployment scenarios**. Being training-free facilitates fast, low-resource compression, and allows for easy adaptation to various models. We believe this is the reason why GPTQ and AWQ are the only weight compression methods well integrated into commonly used LLM serving engines like vLLM[1] and LMDeploy[2]. **Most recent quantization-based methods with stronger performance incorporate further fine-tuning** to maintain performance at high compression ratios [3][4][5]. On the other hand, **GPTQ and AWQ are still considered very strong baselines in the training-free setting**, and recent training-free approaches still perform the comparison with these baselines, for example, DAQ(Oct 2024, [6]) and GWQ(Nov 2024, [7]).
> 2. **Bridging the gap between decomposition-based method and quantization-based method**: As you depicted in “Strength”, BitStack is the first decomposition-based method that achieves comparable performance to quantization approaches. We consider this a huge step as **previous decomposition-based methods usually compare their performance with plain SVD on LLMs**[8][9][10], which leads to significant performance degradation, and can not reach high compression ratios like quantization-based methods (as we depicted in Section 3.1.1). So **we choose to compare with practical quantization-based methods to verify the potential of BitStack in practical deployment** instead of just comparing to other decomposition-based methods. Nevertheless, we reproduce the results of ASVD[8] and SVD-LLM[10], and compare the perplexity of the Llama 2 7B model on the WikiText2 test set at different compression ratios:
>
> |  | **0.1** | **0.2** | **0.3** | **0.4** | **0.5** |
> | --- | --- | --- | --- | --- | --- |
> | ASVD | 5.89 | 9.88 | NaN | NaN | NaN |
> | SVD-LLM | 7.27 | 8.38 | 10.67 | 16.14 | 33.28 |
> | SVD-LLM (with param update) | 7.60 | 8.84 | 11.15 | 16.11 | 27.20 |
> | BitStack | 5.74 | 5.75 | 5.78 | 5.79 | 5.85 |
> |  | **0.6** | **0.7** | **0.8** | **0.9** |  |
> | ASVD | NaN | NaN | NaN | NaN |  |
> | SVD-LLM | 89.92 | 253.22 | 570.56 | 1474.09 |  |
> | SVD-LLM(with param update) | 54.19 | 125.17 | 356.43 | 966.83 |  |
> | BitStack | 5.92 | 6.26 | 8.23 | - |  |
>
> It can be seen from the table that **BitStack outperforms previous decomposition-based methods by a significant margin**. Furthermore, it is important to note that these SVD-based baselines compute the compression ratio differently from those of BitStack.
>
> Specifically, the compression ratio in SVD-based methods is derived from the concatenated ratio of singular values, while BitStack’s compression ratio is calculated as (total_memory-compressed_model_memory)/total_memory. This calculation takes into account the memory consumption of uncompressed weights in layers such as embedding and layer norm, which is not considered in the SVD-based methods. For example, an SVD-LLM checkpoint at 0.8 compression ratio consumes 5.44GiB memory while BitStack consumes only 2.51GiB. Given these differences, this is even an unfair comparison for BitStack, further demonstrating the effectiveness of BitStack. We have added these evaluations in the revision of the paper(Section A.6).

---

> > ### Author Response · Authors · 2024-11-18
> > **Response to Reviewer g6Mj (2/3)**
> >
> > > **Q2**:  The performance gap between BitStack and AWQ is relatively small.
> > >
> >
> > **A2**: Thank you for pointing this out. In certain cases, yes, the performance gap is not significant to quantization baselines, which **well aligns with our claim: be comparable or surpass these baselines.** However, we would like to clarify that **BitStack is designed and optimized for its flexibility, rather than maximizing performance**. As discussed in the introduction, quantization-based methods like AWQ suffer from inflexibility due to their predefined compression ratios, which limit their adaptability in variable memory environments. In contrast, **BitStack excels in such environments due to its ability to dynamically adjust to different memory constraints.** As stated in Section 2.1.1, we deliberately omit certain tricks for higher performance (such as grid search for scales, weight clipping, etc.) to rather keep the overall approach simple. **Our primary contribution is in the flexibility of BitStack, which serves as a practical alternative to traditional baselines in scenarios where the available memory is variable.** That being said, **there are also cases where BitStack significantly outperforms AWQ.**  For instance, in terms of average accuracy scores on LLaMA 3.1 models, BitStack surpasses AWQ-w2g128 by 13.7 points (42.4 vs 28.7) with the 8B model and by 41.3 points (71.1 vs 29.8) with the 70B model. This trend is also observed with both LLaMA 2 and LLaMA 3 models.
> >
> > > **Q3**: When the compression ratio is below 70%, BitStack's performance tends to be weaker than AWQ on most benchmarks.
> > >
> >
> > **A3**: Thank you for pointing this out. **The performance gap between BitStack and AWQ below 70% compression ratio is rather negligible considering that BitStack can significantly outperform AWQ  at higher compression ratios.** For instance, as shown in Table 2, the difference in average accuracy between AWQ and BitStack (Acc(AWQ) - Acc(BitStack)) is **no more than 1.7, 1.3, and 0.1** for the 7B, 13B, and 70B LLaMA 2 models, respectively. However, the gap where BitStack outperforms AWQ (Acc(BitStack) - Acc(AWQ)) reaches **up to 23.6, 30.3, and 42.0** at higher compression ratios. This indicates that BitStack maintains comparable performance to the baselines at lower compression ratios across all scales, while significantly outperforming them at higher compression ratios.

---

> > > ### Author Response · Authors · 2024-11-18
> > > **Response to Reviewer g6Mj (3/3)**
> > >
> > > > **Q4**: What is the real-world inference speed of BitStack? This is a crucial aspect for a compression method targeting large language models (LLMs), yet it is not addressed in the paper.
> > > >
> > >
> > > **A4**:  We agree that inference speed is a crucial part of compression methods, and we included a section analyzing the inference overhead in the initial submission of the paper(previously Section A.5, Section A.5.1 in the revision). **To further address your concerns, we have implemented inference kernels with OpenAI Triton (naively) and already gained a speed boost from 3x to 10x compared to the original PyTorch implementation.** Please refer to Section A.5.2 in the revision of the paper for more details.
> > >
> > > Additionally, **we have measured the throughput of BitStack and compared it with other techniques like memory offloading**. Detailed results and analyses are provided in the revision of the paper (Section A.5.2, see also our response to Reviewer 9SjC). We hope our efforts can address your concerns about the real-world inference speed of BitStack.
> > >
> > > > **Q5**: The font size in Figures 1, 2, 5, 6, and 7 is too small.
> > > >
> > >
> > > **A5**: We appreciate your suggestion on the readability of the paper. We have adjusted the font size in the revision of the paper.
> > >
> > > We hope the responses above have addressed your concerns. Please feel free to raise any further questions if anything remains unclear. We look forward to your reply! :)
> > >
> > > [1] https://github.com/vllm-project/vllm
> > >
> > > [2] https://github.com/InternLM/lmdeploy
> > >
> > > [3] Extreme Compression of Large Language Models via Additive Quantization https://arxiv.org/abs/2401.06118
> > >
> > > [4] QuIP#: Even Better LLM Quantization with Hadamard Incoherence and Lattice Codebooks https://arxiv.org/abs/2402.04396
> > >
> > > [5] QTIP: Quantization with Trellises and Incoherence Processing https://arxiv.org/abs/2406.11235
> > >
> > > [6] DAQ: Density-Aware Post-Training Weight-Only Quantization For LLMs https://arxiv.org/abs/2410.12187
> > >
> > > [7] GWQ: Gradient-Aware Weight Quantization for Large Language Models https://arxiv.org/abs/2411.00850
> > >
> > > [8] ASVD: Activation-aware Singular Value Decomposition for Compressing Large Language Models https://arxiv.org/abs/2312.05821
> > >
> > > [9] Language model compression with weighted low-rank factorization https://arxiv.org/abs/2207.00112
> > >
> > > [10] SVD-LLM: Truncation-aware Singular Value Decomposition for Large Language Model Compression https://arxiv.org/abs/2403.07378

---

> > > > ### Comment · Reviewer_g6Mj · 2024-11-22
> > > >
> > > > Thank the authors' detailed response. Regarding Q5, is there a latency comparison between BitStack and quantization-based methods?

---

> > > > > ### Author Response · Authors · 2024-11-23
> > > > >
> > > > > Thank your for your response. We have further conducted evaluations on the latency between BitStack and GPTQ to address this question. For GPTQ, we use the well optimized inference kernels in [11] to measure the practical inference latency. We measure the latency of generating 50 tokens on an H100 with Llama 2 7B at different compression settings and align the BitStack model to the same compression ratio, the results are as following:
> > > > >
> > > > > |  | w2 | w2g128 | w3 | w3g128 | w4 | w4g128 |
> > > > > | --- | --- | --- | --- | --- | --- | --- |
> > > > > | GPTQ | 1.58 | 1.78 | 3.99 | 4.31 | 1.97 | 2.10 |
> > > > > | BitStack | 1.79 | 1.95 | 2.32 | 2.34 | 2.60 | 2.69 |
> > > > >
> > > > > As shown in the table, the latency difference between BitStack and quantization-based methods is insignificant, especially given that the inference kernels for GPTQ are highly optimized by the community. We are actively working on optimizing the kernels for BitStack to further reduce inference latency in the future.
> > > > >
> > > > > We hope this answers your question regarding latency. Please let us know if there are any other questions!
> > > > >
> > > > > [11] A new refactor of the well adopted repo AutoGPTQ, maintained by the AutoGPTQ’s authors. https://github.com/ModelCloud/GPTQModel

---

> > > > > > ### Comment · Reviewer_g6Mj · 2024-11-25
> > > > > >
> > > > > > Thanks to the authors for the response. I will raise my score as my concerns are addressed.

---

> > > > > > > ### Author Response · Authors · 2024-11-25
> > > > > > >
> > > > > > > Thank you for your thoughtful feedback and for raising the score. We are glad that our response addressed your concerns. We greatly appreciate your time and effort in reviewing and helping to improve our work.

---

### Official Review · Reviewer_9SjC · 2024-11-03

**Soundness:** 3
**Presentation:** 3
**Contribution:** 3
**Rating:** 6
**Confidence:** 3

**Summary:**

This paper introduces BitStack, a novel training-free compression approach for Large Language Models (LLMs) that addresses the challenge of deploying models in environments with variable memory constraints. Unlike traditional compression methods such as quantization that require predefined compression ratios and separate compression processes for each setting, BitStack enables dynamic adjustment of model size at the megabyte level through an innovative weight decomposition technique. The method works by iteratively decomposing weight matrices based on parameter significance, creating approximately 1-bit per parameter residual blocks that are stored and can be loaded flexibly based on available memory. BitStack not only provides fine-grained control over the memory-performance trade-off but also matches or exceeds the performance of established quantization methods like GPTQ and AWQ, particularly at extreme compression ratios. This approach is particularly valuable for local deployment scenarios where available memory may fluctuate due to other applications running concurrently.

**Strengths:**

- Novel Problem Framing: The paper addresses a practical and important problem of deploying LLMs in variable memory environments
It identifies a gap in existing compression methods that typically require fixed compression ratios.
- Practical Utility: Enables dynamic trade-offs between memory usage and model performance eliminating the need for multiple compressed versions of the same model. This reduces storage burden by avoiding separate compression processes for different ratios
- A comprehensive set of experiments across different model scales with convincing results and good ablations to justify different design choices.

**Weaknesses:**

### Major
- This paper is motivated to work in variable memory environments where the amount of memory changes and a compression algorithm is provided. It would be great to have some results on how this compares in efficiency to other memory management techniques like paging. In what scenario is and how more effective is BitStack compared to a systems level optimization that allows to offload parts of the model not currently being used ?

### Minor
- In Table 1, 2,3 it might be a good idea to make the best method's results in bold, to make it easier for the reader.
- Could you clarify what the multiple folders and smaller llama in figure 1a actually refer to ?
- In the abstract the paper references that the primary bottleneck for LLMs is capability and not availability. How does this factor in with other LLM problems like hallucination, knowledge cutoffs etc?

**Questions:**

N/A

---

> ### Author Response · Authors · 2024-11-18
> **Response to Reviewer 9SjC (1/2)**
>
> We greatly appreciate your careful review of our paper and the insightful feedback you've provided. We are glad to see that the novelty and practical utility of BitStack were well recognized. We are grateful for your constructive suggestions, below are our responses to the key comments raised:
>
> > **Q1**: Comparison in efficiency to other memory management techniques.
> >
>
> **A1**: Thank you for this insightful suggestion. We greatly appreciate your constructive feedback, which allows us to enhance the quality of our paper. We agree that comparing BitStack’s efficiency with other system-level memory management techniques would offer valuable context.
> Specifically, while most off-the-shelf LLM serving engines support paging only for KV cache (e.g., vLLM), they do not typically offload model weights. To address this, we compare the efficiency of BitStack with the Hugging Face `accelerate` library, which uses a memory-mapping strategy for offloading model weights [1].
>
> **The main difference between BitStack and weight offloading is, given an available memory budget, BitStack adjusts its compression ratio to fit into the available memory and inference with the compressed model; while memory offloading methods do inference with all model weights and have to load/offload every part of the model during each inference.**
>
> To illustrate this difference, we present a comparison of the throughput (tokens/s) for Llama 2 7B under various memory budgets. Notably, Llama 2 7B requires approximately 12.55GiB of memory to load fully.
>
> |  | **2GiB** | **3GiB** | **4GiB** | **5GiB** | **6GiB** | **7GiB** |
> | --- | --- | --- | --- | --- | --- | --- |
> | BitStack | 21.54 | 21.84 | 17.43 | 14.48 | 12.40 | 10.84 |
> | Offload | 0.35 | 0.37 | 0.42 | 0.48 | 0.52 | 0.57 |
> |  | **8GiB** | **9GiB** | **10GiB** | **11GiB** | **12GiB** | **13GiB** |
> | BitStack | 9.60 | 8,65 | 7.80 | 7.17 | 6.63 | 6.13 |
> | Offload | 0.68 | 0.85 | 1.21 | 2.00 | 4.19 | 27.20 |
>
> As the table shows, BitStack consistently outperforms model offloading in scenarios where available memory is less than 12.55GiB (i.e., when offloading is required). **The throughput of BitStack is higher by a factor ranging from 1.6x to 61.4x, depending on the memory budget.** This demonstrates that BitStack provides a more efficient solution for memory-constrained environments, as it eliminates the need for frequent model weight offloading, which would otherwise incur significant performance penalties. Note that these results were measured on a node with H800 GPUs, which offer high memory bandwidth. The latency caused by loading and unloading model weights could be even more pronounced on edge devices with lower memory bandwidths, further emphasizing the advantages of BitStack in memory-constrained environments.
>
> We have incorporated these results into Section A.5.2 in the revised paper. Once again, we appreciate your valuable suggestion!
>
> > **Q2**: In Table 1, 2,3 it might be a good idea to make the best method's results in bold, to make it easier for the reader.
> >
>
> **A2**: We appreciate your suggestion to highlight the best results in Tables 1, 2, and 3. We have updated the tables in the revision of the paper.

---

> > ### Author Response · Authors · 2024-11-18
> > **Response to Reviewer 9SjC (2/2)**
> >
> > > **Q3**: Could you clarify what the multiple folders and smaller llama in figure 1a actually refer to ?
> > >
> >
> > **A3**: Apologies for the confusion. In Figure 1a, the screen represents the running memory of the device, while the llama displayed on the screen symbolizes the BitStack model. The folders represent other running applications on the device, which also consume memory.  The overall idea conveyed by Figure 1a is that a BitStack model can dynamically adjust its size to fit the available memory, which may fluctuate depending on how much memory is being used by other processes.
> >
> > > **Q4**: In the abstract the paper references that the primary bottleneck for LLMs is capability and not availability. How does this factor in with other LLM problems like hallucination, knowledge cutoffs etc?
> > >
> >
> > **A4**: Apologies for the confusion. What we meant in the abstract is that the bottleneck has shifted **from capability to availability**. **The bottleneck here refers to the challenge of "making the powerful capabilities of large language models accessible to a broader range of users.”** While issues like hallucinations and knowledge cutoffs are typically tied to model limitations (such as training data and architecture), larger models generally offer improvements in handling complex reasoning and generating more accurate responses. However, scaling up these models also makes deployment more challenging, especially on personal devices with limited memory and computational power. Therefore, BitStack (along with other model compression methods) aims to address this challenge by enabling the deployment of larger models in memory-constrained environments.
> >
> > We hope the responses above have addressed your concerns. Please feel free to raise any further questions if anything remains unclear. We look forward to your reply! :)
> >
> > [1] https://huggingface.co/docs/accelerate/en/package_reference/big_modeling#accelerate.disk_offload

---

> > > ### Comment · Reviewer_9SjC · 2024-11-22
> > >
> > > Thanks to the authors for addressing all questions. I will keep my score.

---

> > > > ### Author Response · Authors · 2024-11-23
> > > >
> > > > Thank you for your response. We are grateful for your time and effort in reviewing our work. Have the revisions and new experiments met your expectations, or are there any further questions or suggestions? We would be happy to address any follow-up questions!

---

### Official Review · Reviewer_HY8o · 2024-11-04

**Soundness:** 3
**Presentation:** 4
**Contribution:** 3
**Rating:** 8
**Confidence:** 3

**Summary:**

Authors propose BitStack, a training-free weight compression method using weight decomposition. The method performs well on extreme compression ratio. The method also allows model of an arbitrary size for whichever the GPU memory allows.

**Strengths:**

1. Paper is clearly written and well-motivated.
2. A method that adapt models to different sizes dynamically should be very useful in practice.

**Weaknesses:**

See questions

**Questions:**

1. How long does it take to dynamically load/off-load a residual block? What is the strategy to achieve the best performance/latency tradeoff? Is more or less adjustment preferred during inference?
2. Based on the comparison between 8B and 70B, it seems that one should use llama-8B at FP16 instead of llama-70B at an extreme compression ratio. At least in this case, the extreme quantization does not seem to be needed. And I assume that the latency of a quantized larger model is also worse than the latency of a smaller model at FP16.

---

> ### Author Response · Authors · 2024-11-18
> **Response to reviewer HY8o**
>
> Thank you for your detailed and thoughtful feedback on our paper. We appreciate your positive comments on the clarity of the paper and the practical usefulness of our method, BitStack. We also value your constructive questions, and we would like to address them as follows:
>
> > **Q1.A**:  How long does it take to dynamically load/off-load a residual block?
> >
>
> **A1.A**: The time required to load or off-load a residual block depends on the specific hardware configuration. However, we decompose the weights into residual blocks with minimal sizes (as shown in Section A.4, Table 6). For example, the maximum size of a residual block in an 8B model is 7.56 **megabytes**. In contrast, the memory bandwidth of current devices is measured in **gigabytes per second** (e.g., 8GiB/s for the Snapdragon 8 Gen 2 and 6.4GiB/s for the Apple A17 Pro [1]). Therefore, the time taken to load or off-load a residual block should be **negligible**, especially when compared to the time required to load an entire quantized model at a difference compression ratio, which operates at the gigabyte scale.
>
> > **Q1.B**: What is the strategy to achieve the best performance/latency tradeoff?
> >
>
> **A1.B**: BitStack mainly implements a trade-off between memory and model performance(quality), **we recommend using as much available memory as possible to ensure the best possible model quality**, leveraging the flexibility of BitStack. If latency is taken into account, since the BitStack model incurs higher latency when using more memory (more residual blocks are unpacked on the fly), we **recommend setting a memory range for BitStack models based on the task difficulty in real-world scenarios**. For instance, when handling simple tasks like text summarization, limit the model to using less memory to achieve lower latency; while for more complex tasks, such as complex reasoning, allow the model to use more memory for better performance. For more details on the latency/throughput of BitStack, please refer to Section A.5 in the revised version of the paper.
>
> > **Q1.C**: Is more or less adjustment preferred during inference?
> >
>
> **A1.C**: Adjustment during inference is **encouraged**, as the **flexibility** of BitStack is what sets it apart from other methods. By making dynamic adjustments, we can **maximize the utilization of available memory**, which ultimately leads to improved model performance.
>
> > **Q2**: Based on the comparison between 8B and 70B, it seems that one should use llama-8B at FP16 instead of llama-70B at an extreme compression ratio. At least in this case, the extreme quantization does not seem to be needed. And I assume that the latency of a quantized larger model is also worse than the latency of a smaller model at FP16.
> >
>
> **A2**: You are correct that the 8B model at FP16 outperforms the 70B model at extreme compression ratios with lower latency. In fact, **this holds true for most weight compression methods**, including the baselines discussed in the paper, as well as other well-established methods such as OmniQuant[2], despite it not being training-free.
> You can refer to [3], which provides comprehensive empirical results comparing a wide range of quantization methods, all of which face this issue, referred to as *Pareto Optimal* in the AQLM paper[4].
> Some recent state-of-the-art quantization methods report higher performance for extremely compressed larger models compared to smaller models at full precision. However, **these methods all involve further fine-tuning after compression** and complicate vector quantization[4][5][6] **at each predefined compression ratio**, whereas BitStack is training-free and only requires performing compression once.
> **One typical situation where one would use a 70B BitStack model over an 8B one is when the total memory is relatively large**,  allowing the 70B model to fully leverage its potential when available memory is sufficient.
>
> We hope the responses above have addressed your concerns. Please feel free to raise any further questions if anything remains unclear. We look forward to your reply! :)
>
> [1] https://nanoreview.net/en/soc-compare/qualcomm-snapdragon-8-gen-2-vs-apple-a17-pro
>
> [2] OmniQuant: Omnidirectionally Calibrated Quantization for Large Language Models https://arxiv.org/abs/2308.13137
>
> [3] An Empirical Study of LLaMA3 Quantization: From LLMs to MLLMs https://arxiv.org/abs/2404.14047v2
>
> [4] Extreme Compression of Large Language Models via Additive Quantization ****https://arxiv.org/abs/2401.06118
>
> [5] QuIP#: Even Better LLM Quantization with Hadamard Incoherence and Lattice Codebooks https://arxiv.org/abs/2402.04396
>
> [6] QTIP: Quantization with Trellises and Incoherence Processing https://arxiv.org/abs/2406.11235

---

> > ### Comment · Reviewer_HY8o · 2024-11-25
> > **Clarifications on question 1**
> >
> > My question 1 refers to the frequency of loading/off-loading of a residual block under the circumstance where available memory changes frequently. For example, if another application has drastically changing memory every second. Since it does not make sense to change model size at every milisecond (though the available memory might change at this rate), I am specifically asking about the performance/latency tradeoff in this case. (i.e. more loading/off-loading gives better performance yet induces more overhead in loading/offloading).
> >
> > For question 2, I agree this question is also common for other weight compression methods. But since the paper's strength is on a realistic use case (and currently there is no reason for anyone to use an extremely quantized 70B model instead of an 8B FP16 model), the better performance at extreme compression ratio seems to be less of a highlight here.

---

> > > ### Author Response · Authors · 2024-11-26
> > >
> > > Thank you for the clarifications.
> > >
> > > **Answer to Q1**
> > >
> > > Regarding Q1, we believe the currently available memory plays an important role in this case, as the tradeoff between memory and performance is not linear (as shown in Figure 1b). This is because we perform decomposition on the residuals of previous approximations, meaning the earlier “residual blocks” are more important than the later ones in each weight stack. When the available memory is low, and the model is operating at a high compression ratio, loading a limited number of blocks can result in a significant performance improvement. For example, Llama 3.1 70B achieves an average performance boost of 40 absolute points when moving from an 85% compression ratio to 83%, but gains only 1.5 points when shifting from 79% to 77%, despite loading the same additional number of blocks (Table 1). Therefore, we recommend setting a series of memory threshold points that trigger loading/offloading, with denser intervals at low available memory and sparser intervals at high available memory(assuming the cost of loading blocks is similar under different memory conditions).
> > >
> > > **Answer to Q2**
> > >
> > > For Q2, we believe that a key advantage distinguishing BitStack from other weight compression methods is its one-time compression process, eliminating the need to switch models as memory conditions vary.
> > >
> > > Allow us to be more specific where one would be using a BitStack 70B model over a smaller FP16 model: for instance, an FP16 Llama 2 7B model consumes 12,852 MB of memory with an average zero-shot performance of 69.8, whereas an 85% compressed BitStack Llama 2 70B model consumes 19,363 MB of memory while achieving an average zero-shot performance of 72.2 (Table 2). A 70B model is particularly useful when the available memory on your device is greater than 20000MB most of the time(e.g. on a MacBook Pro 32/64 GB), and you want your model to perform more complex tasks, such as reasoning, although the inference time would indeed be longer.

---

> > > > ### Comment · Reviewer_HY8o · 2024-11-27
> > > >
> > > > Thanks authors for these responses. Overall I like this interesting setting where memory varies overtime. I am glad to raise my score

---

> > > > > ### Author Response · Authors · 2024-11-27
> > > > >
> > > > > Thank you for acknowledging our work! We sincerely appreciate the time and effort you've put into reviewing our paper and helping us improve.

---

### Author Response · Authors · 2024-12-04
**General Response**

We sincerely appreciate the thoughtful and constructive feedback provided by all the reviewers. Your insights have been instrumental in enhancing the quality of our work.

We are glad that the reviewers find our paper:

- Clearly written (`HY8o`);
- Addressing a crucial problem / Well-motivated (`9SjC`, `g6Mj`, `GmCn`);
- Useful in practice(deployment) (`HY8o`, `9SjC`, `g6Mj`);
- Backed with solid experiments (`9SjC`, `GmCn`).

Below is a brief summary of the revisions addressing the key concerns raised by the reviewers:

- **Comparison with system-level memory management techniques (Section A.5.2).** We have performed additional experiments comparing BitStack with the widely adopted memory offloading method. Experiments (Figure 11) show that BitStack provides a speedup over model offloading by a factor of 1.6x to 61.4x across different memory budgets, while maintaining comparable quality at most memory budgets.
- **Inference speed (Section A.5.2).** To enhance the practicality of BitStack, we further implemented inference kernels using OpenAI Triton and compared the efficiency of different implementation strategies. As a result, we achieved a speedup over the original PyTorch implementation by a factor of 3x to 10x based on the memory budget, drastically improving the inference efficiency (Figure 10). We further conducted experiments to compare the inference speed of BitStack with a quantization-based method (GPTQModel, a widely adopted implementation of GPTQ with fused inference kernels), and show that the inference speed is comparable with it. We sure will keep optimizing these inference kernels for the best practicality of BitStack, and everything will be open sourced.
- **Comparison with other decomposition-based weight compression methods (Section A.6).** BitStack is the first training-free method that bridges the performance gap between decomposition-based methods and quantization-based methods. To justify this, we added additional experiments to compare the efficacy of BitStack with other decomposition-based methods like ASVD and SVD-LLM. Results show that BitStack significantly outperforms these methods, even with much less memory usage (Table 7).

We would like to express our gratitude again for all the effort put by the reviewer in improving our work. Your valuable insights, constructive feedback, and thoughtful suggestions have significantly enhanced the quality and clarity of our manuscript. Thank you for your invaluable contributions.

---

### Meta-Review · Area_Chair_NQhp · 2024-12-15

**Metareview:**

This paper studies a compression method, BitStack, that allows dynamic adjustment of the weight matrix size based on available memory, achieving a balance between memory usage and performance. When there is available memory, one will retrieve pre-sorted weight blocks into the memory. The decomposition is based on a dynamic SVD process applied to a scaled version of the weight matrix.

Main strengths:
- A very useful technique for dynamic adjustment of weight matrix sizes.
- Strong performance compared to other compression methods.

Main weaknesses:
- There needs to be more discussion on system-level designs.
- IMHO, the paper does not thoroughly discuss related work. For example, machines may be preempted dynamically in elastic computing, and many papers study how to adjust the computational load during computation.

**Additional Comments On Reviewer Discussion:**

During the rebuttal, the authors studied more system-level memory management techniques (such as memory offloading) and compared them to the BitStack dynamic adjustment method. They also provided evaluations on the latency of BitStack.

---

### Decision · Program_Chairs · 2025-01-22

Accept (Poster)